# MetaDNS: Enhancing Exploration in Discrete Neural Samplers via Well-Tempered Metadynamics

**Xiaochen Du** [1]   **Juno Nam** [1]   **Jaemoo Choi** [2]   **Wei Guo** [2]   **Sathya Edamadaka** [1]   **Junyi Sha** [1]   **Elton Pan** [1]
**Yongxin Chen** [2]   **Molei Tao** [2]   **Rafael Gómez-Bombarelli** [1]

## Abstract

Sampling from discrete distributions with multiple modes and energy barriers is fundamental to machine learning and computational physics. Recent discrete neural samplers like MDNS suffer from mode collapse and fail to sample high-energy barrier regions between modes, which is critical for free energy estimation and understanding phase transitions. We propose Metadynamics Discrete Neural Sampler (MetaDNS), a general framework integrating well-tempered metadynamics into discrete diffusion or autoregressive samplers. By maintaining an adaptive, history-dependent bias potential along selected low-dimensional coordinates, MetaDNS forces exploration of previously inaccessible regions, enabling free energy reconstruction infeasible with standard neural samplers due to a lack of high-energy samples. On challenging low-temperature benchmarks including Ising, Potts, and the copper-gold binary alloy, MetaDNS reproduces the thermodynamic distribution. Compared to MCMC-based metadynamics, MetaDNS also achieves comparable exploration requiring fewer bias deposition steps.

## 1. Introduction

Predicting equilibrium properties of crystalline materials, such as phase stability in alloys or magnetic ordering, often relies on sampling Boltzmann distributions defined over discrete configurational spaces, typically arising from cluster expansion or effective spin Hamiltonians on a fixed lattice. Specifically, these are distributions $\pi(x) \propto e^{-\beta E(x)}$, where the energy function $E(x)$ encodes interactions between dis-

crete degrees of freedom (atomic occupancies, spins, etc.) on a lattice and $\beta = 1/k_{\mathrm{B}}T$ is the inverse temperature, commonly used in statistical physics and materials science (Van der Ven et al., 2018; Ångqvist et al., 2019; Chang et al., 2019). Markov-chain Monte Carlo (MCMC) methods, such as the Metropolis–Hastings (MH) algorithm (Metropolis et al., 1953) or Glauber dynamics (Glauber, 1963; Süzen, 2014), have historically served as the conventional solutions for these settings. However, they suffer from critical slowing down near phase transitions or in rugged energy landscapes with many local optima. In these multimodal settings, local samplers struggle to traverse such high-energy barriers, leading to slow mixing and biased estimation of physical observables (Faulkner & Livingstone, 2024).

To overcome the limitations of local sampling, recent advances have formulated discrete sampling from Boltzmann distributions as a generative modeling problem. Unlike amortized generative models (e.g., diffusion or autoregressive networks) trained on data via maximum likelihood to approximate an empirical distribution (Song et al., 2021; Lou et al., 2024), *neural samplers* for Boltzmann targets take as input only the energy function $E(x)$ and learn to transform a simple reference distribution (e.g., uniform or masked) into the target $\pi(x)$ without requiring pre-existing samples from the target (Liu et al., 2024; Holderrieth et al., 2025; Zhu et al., 2025). This line of work has shown promise in scaling to high-dimensional discrete spaces where traditional MCMC struggles.

Despite these theoretical strides, discrete neural samplers remain vulnerable to mode collapse. When trained via variational objectives (e.g., minimizing Kullback–Leibler (KL) divergences), state-of-the-art methods such as MDNS (Zhu et al., 2025) tend to concentrate probability mass on modes discovered early in training, failing to traverse low-probability bottlenecks to explore thermodynamically relevant but separated states. This failure has two critical consequences. First, these models miss entire modes in multimodal distributions, yielding biased estimates of equilibrium properties. Second, and more subtly, they fail to generate samples in high-energy regions between modes, precisely the thermodynamically critical barrier-crossing

---

[1]Massachusetts Institute of Technology [2]Georgia Institute of Technology. Correspondence to: Rafael Gómez-Bombarelli <rafagb@mit.edu>.

*Proceedings of the 43rd International Conference on Machine Learning*, Seoul, South Korea. PMLR 306, 2026. Copyright 2026 by the author(s).

configurations needed to understand transition pathways and estimate free energy landscapes. Recent efforts such as Proximal Diffusion Neural Sampler (PDNS) (Guo et al., 2026a) introduce iterative proximal steps to prevent collapse, yet still lack explicit mechanisms to force exploration of these high-energy regions.

This limitation becomes particularly severe in realistic materials systems, where evaluating $E(x)$ is computationally expensive. First-principles quantum mechanical calculations (e.g., density functional theory) can require minutes to hours per configuration, while state-of-the-art machine-learned force fields (MLFFs) with millions of parameters require computational time that quickly build up when evaluating over many configurations, common in modern screening and sampling workflows. In this regime, minimizing the number of energy evaluations becomes paramount; a single wasted evaluation on a known low-energy configuration is a missed opportunity to explore other intermediate or local minima states.

In this work, we propose Metadynamics Discrete Neural Sampler (MetaDNS), a general framework that actively combats mode collapse and enables exploration of the Boltzmann distribution by integrating well-tempered metadynamics (WT-MetaD) into the training of discrete neural samplers. MetaDNS is agnostic to the type of discrete neural sampler, whether formulated via CTMCs (Holderrieth et al., 2025; Zhu et al., 2025) or order-agnostic autoregressive models (Liu et al., 2024; Ou et al., 2025a). Drawing inspiration from enhanced sampling in molecular dynamics, MetaDNS constructs an adaptive, history-dependent bias potential defined over low-dimensional collective variables (CVs). This bias acts as an intrinsic motivation signal, "filling in" explored energy wells and effectively flattening the landscape during training; by modifying the target path measure on-the-fly, MetaDNS forces the generative model to explore regions that are thermodynamically inaccessible under the unbiased energy. Crucially, MetaDNS retains asymptotically exact sampling from the target Boltzmann distribution through importance reweighting by the bias potential.

As a further benefit, MetaDNS also improves sampling efficiency compared to traditional MCMC-based WT-MetaD approaches. By leveraging neural sampling to generate independent configurations, rather than requiring sequential MCMC chains at each biased energy landscape, MetaDNS achieves comparable or superior exploration with up to $2\times$ fewer bias deposition steps in the Potts model and copper-gold binary alloy system during training. Additionally, the learned bias potential enables estimation of free energy differences along collective variables, providing diagnostic capabilities for understanding the thermodynamic landscape.

We validate MetaDNS on complex discrete benchmarks where state-of-the-art baselines struggle. Beyond standard Ising and Potts models, we introduce the binary alloy copper-gold (Cu-Au) system as a rigorous benchmark for the machine learning community. Unlike simple spin glasses, Cu-Au exhibits complex order-disorder phase transitions and multiple stable intermetallic phases, providing a realistic testbed for materials thermodynamics.

**Contributions.** Our contributions are four-fold: (1) recovery of diverse modes in low-temperature settings with asymptotically exact sampling via importance reweighting; (2) exploration of high-energy regions and free energy landscape reconstruction where standard neural samplers fail; (3) comparable or superior exploration vs. MCMC-based WT-MetaD with $2\times$ fewer bias deposition steps; and (4) a training objective compatible with both diffusion and autoregressive backbones, and the Cu-Au binary alloy as a rigorous benchmark.

## 2. Related Work

### 2.1. Discrete Neural Samplers

Since the seminal work on Boltzmann Generators (Noé et al., 2019), neural samplers have been developed for statistical inference over Boltzmann distributions defined by physical energy functions. Early approaches focused on minimizing the reverse KL divergence using exact likelihood models, including autoregressive models for discrete settings (Wu et al., 2019), and were later applied to sampling on alloy material systems (Damewood et al., 2022). This line of work was extended to any-order autoregressive models through learning general marginal distributions (Liu et al., 2024), and further scaled to larger lattices via architectural improvements (Du et al., 2026).

More recently, advances in continuous and discrete diffusion models (Song et al., 2021; Lou et al., 2024) have motivated discrete diffusion samplers based on continuous-time Markov chain (CTMC) formulations. MDNS (Zhu et al., 2025) casts discrete sampling as aligning path measures of CTMCs and derives training objectives grounded in stochastic optimal control (Zhang & Chen, 2022); it further proposes a weighted denoising cross-entropy (WDCE) loss to scale score-learning-like objectives via importance sampling. PDNS (Guo et al., 2026a) diagnoses mode collapse as a global-optimization pathology and mitigates it by applying proximal point iterations on the space of path measures, instantiating each proximal step with a proximal WDCE objective. Concurrently, Guo et al. (2026b) extend the adjoint Schrödinger bridge sampler (Liu et al., 2025) framework to discrete CTMCs by identifying a cyclic group structure on the state space that enables adjoint matching, achieving competitive sample quality with significant advantages in training efficiency.

LEAPS (Holderrieth et al., 2025) learns a CTMC rate matrix to transport from an easy base distribution to a target distribution and can be viewed as a continuous-time analogue of annealed importance sampling, and it introduces locally equivariant network parameterizations to make rate matrix learning and weight computation tractable in high dimensions. DNFS (Ou et al., 2025b) extends this by estimating the gradient of the normalizing constant rather than parametrizing it, and by introducing a transformer-based architecture for the rate matrix. Finally, TCSIS (Kholkin et al., 2025) introduces the target concrete score identity to estimate the concrete score required for the time reversal of CTMC from the expectation of Boltzmann weights under the forward noising kernel.

These methods substantially advance learning-based discrete sampling, but their exploration mechanisms are primarily driven by convergence of an initial prior distribution to a fixed target distribution during training or depends on the chosen annealing paths, which are not guaranteed to be optimal for mode discovery and barrier crossing. In contrast, our work introduces an explicit *history-dependent* exploration bias in a low-dimensional CV space (metadynamics), targeting mode discovery and barrier crossing in a controllable and interpretable way.

### 2.2. Metadynamics and Enhanced Sampling

Metadynamics is a classical enhanced sampling technique that constructs a history-dependent bias potential $V(s)$ along a collective variable (CV) $s = \xi(x)$ to discourage revisiting already-explored regions and to promote barrier crossing (Laio & Parrinello, 2002). In the well-tempered variant, the bias is tempered so that the CV marginal approaches a softened target, yielding an asymptotic relation $V^\star(s) = -(1 - 1/\gamma)F(s) + c$ with the free energy $F(s)$ (Barducci et al., 2008).

### 2.3. From Continuous to Discrete

More broadly, a growing body of work combines neural networks with enhanced sampling in continuous state spaces. Ribera Borrell et al. (2024) combine stochastic optimal control (SOC)-based importance sampling with adaptive metadynamics, approximating the optimal control by a neural network to accelerate rare-event sampling in metastable diffusions. Zhang et al. (2019) propose TALOS, a GAN-style framework that iteratively trains a sampler and discriminator to learn an optimal bias potential and transport plan that lowers free-energy barriers in molecular settings. Zhu et al. (2026) provide a comprehensive review of how machine learning integrates with enhanced sampling through data-driven collective variables, improved biasing schemes, and generative-model-based strategies across biomolecular and catalytic applications in the continuous domain.

Most directly related to our setting, Nam et al. (2026) propose the well-tempered adjoint Schrödinger bridge sampler (WT-ASBS), which augments a continuous diffusion-based sampler with a WT-MetaD-style repulsive CV bias updated online, and uses reweighting to recover Boltzmann statistics. Empirically, this improves mode discovery and enables free energy estimation in challenging molecular benchmarks. MetaDNS takes inspiration from this continuous setting but targets discrete configuration spaces (Ising/Potts/alloy models), where the sampler dynamics, objectives, and convergence intuitions require different technical treatment.

## 3. Method

We now describe MetaDNS (Figure 1 and Algorithm 1), our framework for integrating WT-MetaD with discrete neural samplers. The key idea is to train a neural sampler on a *time-varying* biased distribution that actively discourages revisiting already-explored regions of the state space, forcing the model to traverse high-energy barriers and discover new modes.

### 3.1. MetaDNS: Algorithm Overview

Given a target Boltzmann distribution $\pi(x) \propto e^{-\beta E(x)}$ over discrete configurations $x \in \mathcal{X}$, MetaDNS maintains a bias potential $V_t(s)$ defined over low-dimensional CVs $s = \xi(x)$, where $\xi : \mathcal{X} \to \mathcal{S}$ projects configurations onto a discrete set of bins. The biased distribution at iteration $t$ is

$$\pi_{V_t}(x) \;\propto\; e^{-\beta[E(x)+V_t(\xi(x))]}.$$

MetaDNS alternates between two steps: (1) **Inner loop**: train the neural sampler $q_\theta$ to approximate $\pi_{V_t}$ for a fixed bias $V_t$; (2) **Outer loop**: update the bias $V_t$ by depositing Gaussian-like "hills" in CV space at CVs regions visited by samples from $q_\theta$. As training progresses, the bias accumulates in frequently visited regions, effectively raising their energy and forcing the sampler to explore new modes. The complete procedure is detailed in Algorithm 1.

### 3.2. Collective Variables

CV $\xi(x)$ choice is problem-dependent. For Ising and Potts models, we use magnetization and per-state occupation counts respectively (see Section 4). For the Cu-Au alloy, we use the fraction of gold atoms ($x_{\text{Au}}$). While not strictly necessary, CVs should capture slow modes of the system and distinguish between metastable states.

### 3.3. Well-Tempered Hill Deposition

The bias update uses a kernel $K(s, s')$ (e.g., discrete Gaussian) centered at the visited CV bin $s' = \xi(x_j)$. The well-tempered factor $\exp(-V_{t-1}(s)/(\gamma k_B T))$ ensures that the bias accumulation slows down in already-visited re-

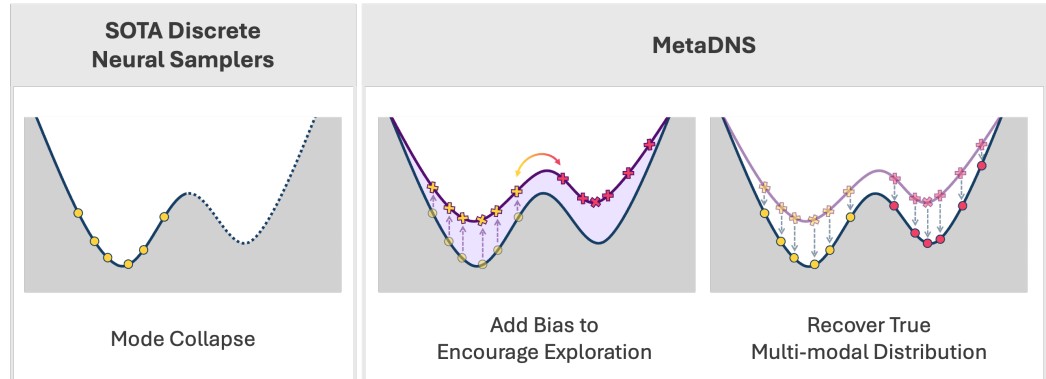

*Figure 1.* Scheme for MetaDNS. (Left) State of the art (SOTA) discrete neural samplers suffer from mode collapse at low temperatures: samples concentrate in a single deep minimum of the multi-modal energy landscape. (Middle) MetaDNS adds a history-dependent bias potential (purple) that raises the energy in frequently visited regions, flattening the landscape and encouraging exploration across energy barriers to visit multiple minima. (Right) Samples drawn from the biased distribution are reweighted to recover the true multi-modal target distribution on the original energy landscape.

gions, preventing the bias from growing indefinitely. At convergence, the bias potential satisfies $V^\star(s) \approx -(1 - 1/\gamma)F(s) + c$, where $F(s)$ is the free energy along the CV and $c$ is a constant. This allows reconstruction of the free energy landscape from the learned bias. From our experiments and in literature (Dama et al., 2014), a Gaussian kernel was more effective than a delta kernel.

### 3.4. Convergence of Neural Metadynamics Sampling

Convergence of metadynamics in ergodic systems (MCMC or molecular dynamics, MD) was established for both continuous and discrete bias potentials, $V_{t-1}(s)$, and CVs, $s$ (Micheletti et al., 2004; Barducci et al., 2008; Crespo et al., 2010; Dama et al., 2014). With a neural sampler, two additional concerns arise: (1) non-ergodicity, since the neural sampler may not satisfy the same ergodicity guarantees as MCMC or MD; and (2) approximation error, if the sampler fails to learn the biased target $E_{\text{biased}}(x) = E(x) + V_{t-1}(\xi(x))$ at each $t$. We discuss these issues, their implications for bias convergence, and mitigations in Section A.

### 3.5. Importance Reweighting for Exact Sampling

Since MetaDNS trains on the biased distribution $\pi_{V_t}$, samples from $q_\theta$ need to be reweighted to recover unbiased estimates of observables $\langle A \rangle_\pi = \sum_x A(x)\pi(x)$ under the original target. We apply self-normalized importance sampling (SNIS) with the choice of importance weights depending on whether the sampler has a tractable likelihood.

**Bias-based Reweighting (for any sampler).** For samplers without exact likelihoods (e.g., uniform discrete diffusion

models), we reweigh using the accumulated bias potential:

$$\langle A \rangle_{\text{bias}} = \frac{\sum_{i=1}^N w_i A(x_i)}{\sum_{i=1}^N w_i}, \quad w_i = \exp(V(\xi(x_i))).$$

This corrects for the bias introduced during training, yielding asymptotically exact estimates as long as the sampler remains close to $\pi_{V_t}$. However, with an imperfect sampler, bias-based reweighting alone can yield biased estimators.

**Likelihood-based Reweighting (for exact-likelihood samplers).** For exact-likelihood samplers, i.e., those where the likelihood $p_\theta(x)$ can be easily computed such as in autoregressive models, we can use likelihood-based importance weights:

$$\langle A \rangle_{\text{likelihood}} = \frac{\sum_{i=1}^N \tilde{w}_i A(x_i)}{\sum_{i=1}^N \tilde{w}_i}, \quad \tilde{w}_i = \frac{\exp(-\beta E(x_i))}{q_\theta(x_i)}.$$

These importance weights yield asymptotically correct estimators of observables (Nicoli et al., 2020; Damewood et al., 2022). In our experiments, bias-based reweighting $w_i = \exp(V(\xi(x_i)))$ is used for global observables (energy, magnetization, CV marginals, and NESS), as it is computationally cheaper (no additional energy evaluations needed) and often has lower variance (weights depend only on low-dimensional CVs). For two-point correlations in Ising and Potts models, likelihood-based reweighting was used for better agreement with the reference method. When exact likelihoods are not available, MetaDNS samples can be used as informed proposals for MCMC correction to preserve statistical exactness (Nicoli et al., 2020) at the cost of additional energy calculations.

**Path likelihood and Radon–Nikodým derivatives (diffusion samplers).** For CTMC-based discrete diffusion samplers, the configuration-level density $q_\theta(x)$ is not directly accessible. However, for MDNS, the autoregressive

---

**Algorithm 1** MetaDNS Training and Inference

1: **Input:** Energy function $E(x)$, inverse temperature $\beta$, CV $\xi(\cdot)$
2: **Hyperparameters:** Bias factor $\gamma > 1$, initial hill height $h$, hill width $\sigma$, inner steps $N_{\text{inner}}$, outer steps $N_{\text{outer}}$, and bias deposition kernel $K(s, s')$
3: **Initialize:** Neural sampler parameters $\theta_0$, bias potential $V_0(s) = 0$ for all $s \in \mathcal{S}$
4:
5: **// Training Phase: Adaptive Bias Construction**
6: **for** $t = 1$ to $N_{\text{outer}}$ **do**
7:    **// Inner Loop: Train sampler on biased distribution**
8:    **for** $k = 1$ to $N_{\text{inner}}$ **do**
9:       Sample $M_{\text{inner}}$ configurations $\{x_i\}_{i=1}^{M_{\text{inner}}}$ from $q_\theta$
10:       Evaluate biased energies: $E_{\text{biased}}(x_i) = E(x_i) + V_{t-1}(\xi(x_i))$
11:       Update sampler: $\theta \leftarrow \theta - \nabla_\theta \mathcal{L}(\theta; \{x_i\}, E_{\text{biased}})$ {e.g., WDCE loss from MDNS}
12:    **end for**
13:
14:    **// Outer Loop: Update bias potential**
15:    Sample $M_{\text{outer}}$ configurations $\{x_j\}_{j=1}^{M_{\text{outer}}}$ from $q_\theta$
16:    **for** each bin $s \in \mathcal{S}$ **do**
17:       $V_t(s) \leftarrow V_{t-1}(s) + \sum_{j=1}^{M_{\text{outer}}} h \exp\left(-\frac{V_{t-1}(s)}{\gamma k_B T}\right) \cdot K(s, \xi(x_j))$ {Well-tempered hill deposition}
18:    **end for**
19: **end for**
20:
21: **// Inference Phase: Generate Samples and Reweight**
22: Sample $N_{\text{inference}}$ configurations $\{x_i\}_{i=1}^{N_{\text{inference}}}$ from final $q_\theta$
23: Compute importance weights: $w_i = \exp(V_{N_{\text{outer}}}(\xi(x_i)))$ for $i = 1, \ldots, N_{\text{inference}}$
24: **Output:** Weighted samples $\{(x_i, w_i)\}_{i=1}^{N_{\text{inference}}}$ for estimating expectations under $\pi(x)$ via SNIS

---

unmasking structure makes the path likelihood tractable, yielding an exact-density interpretation. A path $X = (X_0, X_1, \ldots, X_T)$ is a sequence of configurations over time with final configuration $X_T$. The Radon–Nikodým (RN) derivative $\exp(W^u(X) - \log Z) = d\mathbb{P}^*/d\mathbb{P}^u(X)$ between the optimal path measure $\mathbb{P}^*$ and the learned path measure $\mathbb{P}^u$ defines path-level importance weights directly usable for ESS calculation. Crucially, because MDNS generates samples via autoregressive unmasking, the path measure $\mathbb{P}^u$ factorizes over the unmasking transitions, making the path likelihood tractable. The log-path-likelihood then equals $\log q_\theta(x_T) = \sum_t \log p_\theta(X_t \mid X_{<t})$ by the chain rule, enabling the standard likelihood-based weights $\tilde{w}_i = \exp(-\beta E(x_i))/q_\theta(x_i)$ and making MDNS an exact-density sampler in the same sense as purely autoregressive

models. The full derivation is in Section B.

# 4. Experiments and Results

We evaluate MetaDNS on three benchmark systems of increasing complexity: Ising and Potts models across multiple lattice sizes ($L \in \{4, 8, 16\}$) and inverse temperatures ($\beta$), and the realistic Cu-Au binary alloy system at $2 \times 2 \times 4$ and $4 \times 4 \times 4$ supercells at 500K, 680K, and 1200K. We compare against both MDNS and MCMC-based WT-MetaD across all systems, using Swendsen–Wang (SW) algorithm (Swendsen & Wang, 1987) as ground truth for Ising and Potts models and regular MCMC as ground truth for Cu-Au. Implementation details are in Section F, including the condensed MDNS training pipeline (Algorithm 2) and key hyperparameters (Tables 2 to 4); sensitivity analyses across WT-MetaD hyperparameter ranges are in Figures 17 to 19 (Ising), Figures 20 and 21 (Potts), and Figures 22 and 23 (Cu-Au). We report absolute magnetization (Mag.), average two-point correlation (Corr.), normalized effective sample size (NESS), and Jensen–Shannon (JS) divergence for energy distributions and spin states or atom concentrations. Free energy profiles in Figures 4 and 5 refer to the potential of mean force (PMF) along the chosen collective variable. Formal definitions of these metrics and of the PMF are given in Section C.

## 4.1. Ising Model

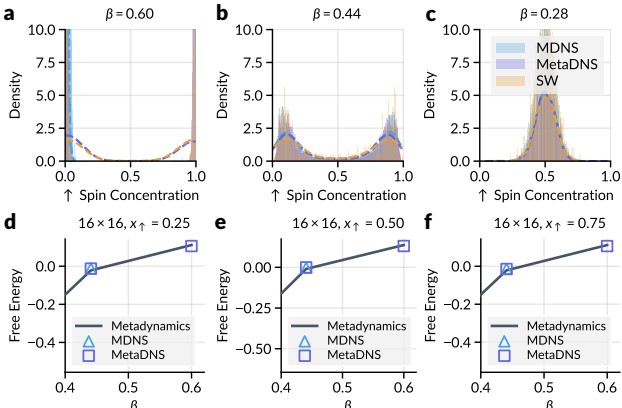

*Figure 2.* Ising model results at $L = 16$: (a-c) Up-spin concentration distributions at three temperatures ($\beta = 0.6, 0.44, 0.28$). MetaDNS (ours) captures the full bimodal distribution at low temperature while MDNS suffers from mode collapse. (d-f) Free energy per site versus inverse temperature for $x_\uparrow = 0.25, 0.5$, and 0.75. Lack of samples at intermediate spin concentrations prevents MDNS from estimating free energies at low temperature.

**Mode discovery and free energy estimation.** We use the up-spin concentration $x_\uparrow$ (fraction of sites with spin $+1$) as the CV, which distinguishes the two magnetized phases and the disordered states in between. Figure 2

*Table 1.* Ising model results at $L = 16$. **Bold** indicates best result. Underlined method name indicates our method (MetaDNS); underlined values indicate SW ground truth. Mag.: absolute magnetization. Corr.: average 2-point correlation. NESS: normalized effective sample size. JS Div.: Jensen-Shannon divergence vs. SW. This table quantifies the distributions in Figure 2; at low temperature, MDNS can appear better on NESS and E. JS Div. because it has collapsed to a narrow mode, whereas the decisive metric for mode coverage is $x_\uparrow$ JS Div., where MetaDNS is substantially better. MDNS warm-started at $\beta_{high}$ for 20k steps before training at $\beta_{low}$ at 30k steps is also included for comparison.

| Inv. Temp. | Method | Mag. | Corr. | NESS ↑ | E. JS Div. ↓ | $x_\uparrow$ JS Div. ↓ |
|---|---|---|---|---|---|---|
| | MDNS | 0.117 | 0.319 | 0.826 | $6.2 \times 10^{-3}$ | $1.7 \times 10^{-2}$ |
| $\beta_{high} = 0.28$ | MetaDNS | **0.120** | **0.321** | **0.850** | $6.9 \times 10^{-3}$ | $1.7 \times 10^{-2}$ |
| | SW (ground truth) | 0.121 | 0.322 | / | / | / |
| | MDNS | **0.712** | **0.727** | **0.871** | $1.1 \times 10^{-2}$ | $3.6 \times 10^{-2}$ |
| $\beta_{crit} = 0.4407$ | MetaDNS | 0.716 | 0.729 | 0.839 | $1.4 \times 10^{-2}$ | $4.2 \times 10^{-2}$ |
| | SW (ground truth) | 0.713 | 0.726 | / | / | / |
| | MDNS | **0.974** | **0.955** | **0.979** | $4.3 \times 10^{-3}$ | $2.2 \times 10^{-1}$ |
| | MDNS (warm-start) | 0.972 | 0.952 | 0.933 | $5.1 \times 10^{-3}$ | $4.8 \times 10^{-3}$ |
| $\beta_{low} = 0.6$ | MetaDNS | **0.974** | **0.955** | 0.426 | $3.3 \times 10^{-2}$ | $4.6 \times 10^{-2}$ |
| | SW (ground truth) | 0.973 | 0.954 | / | / | / |

demonstrates MetaDNS's advantage at $L = 16$. At low temperature (panel a), MDNS suffers from mode collapse, capturing the down-spin phase but missing the up-spin phase, while MetaDNS captures the full bimodal distribution. Table 1 quantifies this gap: at low temperature ($\beta = 0.60$), MetaDNS achieves $\sim 5\times$ lower $x_\uparrow$ JS divergence ($4.6 \times 10^{-2}$ vs $2.2 \times 10^{-1}$) while matching SW ground truth in magnetization and correlation. MetaDNS exhibits lower NESS as it samples a broader bimodal distribution, whereas MDNS's near-unity NESS reflects mode collapse to a narrow unimodal distribution. Visual inspection of sample configurations (Figure 9) confirms this behavior. MDNS generates nearly identical spin configurations, while MetaDNS produces diverse configurations with well-formed domains of both orientations. Similar mode collapse occurs at $L = 8$ (Table 6 and Figure 7), while the smaller $L = 4$ lattice shows no mode collapse at low temperature (Table 5). Doubling the training length (100k steps) does not alleviate the mode collapse problem for MDNS (Figure 9b). Mode collapse is also not limited to this specific temperature, but persists at additional low temperature $\beta > \beta_{crit}$ (Figure 10). At critical ($\beta = 0.4407$) and high ($\beta = 0.28$) temperatures, both methods successfully match the ground truth distribution. Warm-starting MDNS at $\beta_{high}$ before fine-tuning at $\beta_{low} = 0.6$ (Table 1) yields mixed results: $x_\uparrow$ JS divergence improves substantially relative to vanilla MDNS (better phase coverage), but energy JS divergence increases and two-point correlation slightly worsens, so warm-start alleviates but does not resolve mode collapse.

MDNS not only suffers from mode collapse at low temperatures, in addition, since they are sampling from a very narrow distribution range, they do not sample intermediate concentrations ($x_\uparrow = 0.25$ and $0.75$) at this temperature. The lack of samples in these intermediate regions makes free energy estimation impossible using MDNS. By contrast, MetaDNS by virtue of the biased landscape, obtains samples at these intermediate compositions, allowing per-composition free energies (PMF) to be estimated across the full composition range at both low and critical temperatures (panels d-f). Additionally, the free energy profiles from MetaDNS agree well with the WT-MetaD reference (Figures 6 and 7). Two-point correlation functions (Figure 8) validate that after reweighting, MetaDNS maintains correct statistical properties, matching SW ground truth across all lattice sizes and temperatures. This demonstrates that improved exploration does not compromise statistical accuracy.

### 4.2. Potts Model

**Improvements over MCMC-based WT-MetaD.** We use a 2D collective variable (CV 1, CV 2) given by the per-state occupation fractions (e.g., fractions of sites in each of the $q$ Potts states), which distinguish the $q = 3$ ordered phases at the vertices of an equilateral triangle and the disordered configuration near CV $\approx (0, 0)$; see Section D for the explicit definition. The Potts model results demonstrate not only MetaDNS's advantages over MDNS in mode discovery but also its efficiency over MCMC-based WT-MetaD. The collective variable distributions in Figures 3, 11 and 13 further validate MetaDNS's strength at learning all modes: at low temperatures, MDNS collapses to a single mode, while MetaDNS successfully covers all modes and the interspace regions similar to WT-MetaD. When "warm-started" at $\beta_{high}$ (as in (Zhu et al., 2025)), MDNS also fails to overcome mode collapse given the same total training budget (Figure 12, Table 9). See also Figure 15 for a comparison of the MDNS and MetaDNS samples. Correlation profiles and magnetization values (Figure 14 and Tables 7 to 9) confirm that MetaDNS maintains correct statistical properties after reweighting, with strong agreement to SW ground truth.

Compared with MCMC-based WT-MetaD, MetaDNS achieves comparable free energy profiles with significantly fewer bias deposition steps (Figure 4). MetaDNS converges earlier to within 1 $k_B T$ RMSE accuracy, measured with respect to the final WT-MetaD profile at 125k bias deposition steps for each temperature. (Calculation details in Section C.) At low temperature, MetaDNS reaches convergence at 50k steps compared to WT-MetaD's 94.5k steps; at critical temperature, MetaDNS converges at 14k steps versus WT-MetaD's 36k steps; and at high temperature, MetaDNS converges at 40k steps compared to WT-MetaD's 107k steps; see Section F.6 for a per-step energy evaluation breakdown.

This speedup in bias deposition steps is due to a fundamental difference in exploration strategies (Figure 16) between MetaDNS and MCMC-based WT-MetaD. WT-MetaD suf-

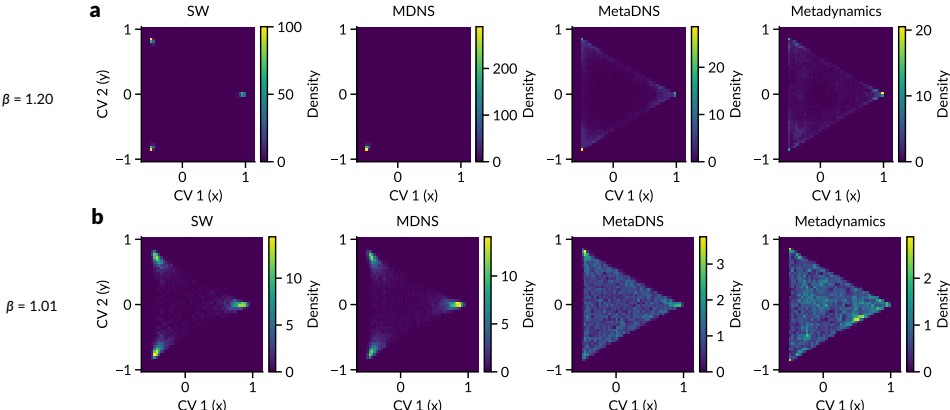

*Figure 3.* Collective variable (CV) distributions for Potts ($q = 3$, $L = 16$) models at two temperatures: (a) low and (b) critical. 2D CV space (CV 1 vs. CV 2) comparing SW (ground truth), MDNS, MetaDNS (ours), and MCMC-based WT-MetaD. At low and critical temperatures, MDNS exhibits mode collapse, discovering only one of three modes. MetaDNS successfully covers all modes and interspace regions, matching WT-MetaD coverage.

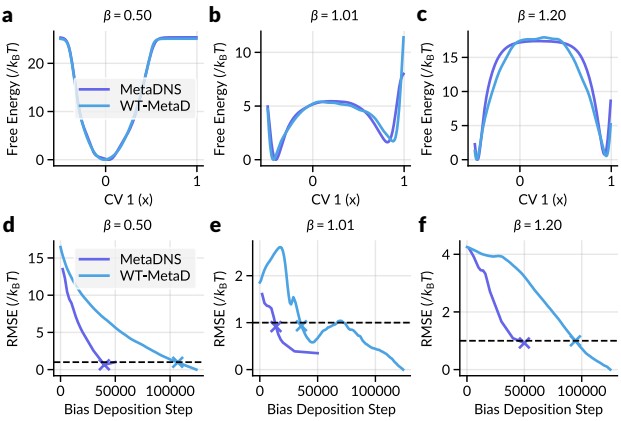

*Figure 4.* Free energy profiles and convergence for Potts ($q = 3$) at $L = 16$ comparing MetaDNS (ours) with MCMC-based WT-MetaD. (a-c) Free energy profiles along CV 1 at high ($\beta = 0.5$), critical ($\beta = 1.01$), and low ($\beta = 1.2$) temperatures showing agreement with WT-MetaD. (d-f) Convergence speedup of MetaDNS over WT-MetaD at the same temperatures (RMSE vs. steps).

fers from high correlation between sequential MC steps as it spends considerable time in random configurations (CV $\approx (0, 0)$) before gradually spreading to the target modes. In contrast, MetaDNS leverages neural sampling to generate independent samples, allowing it to quickly target and discover modes. Although MetaDNS discovers modes sequentially, it eventually covers all modes with fewer bias deposition steps than WT-MetaD. Unlike WT-MetaD, which must re-mix the Markov chain against each updated bias landscape, MetaDNS amortizes sampling to a single forward pass per bias deposition step.

When accounting for wall-clock time, however, the cost of neural network optimization dominates for Potts model that

has a simple analytical energy function, resulting in overall much longer wall-clock training time than WT-MetaD (20 h vs. 1h on an A100 GPU, see Table 12). Nevertheless, generating new configurations is significantly faster with MetaDNS. For the $16 \times 16$ Potts model, MetaDNS generates 10k samples via 256 autoregressive unmasking steps (1 per lattice site) in under 1 min. In contrast, WT-MetaD requires ∼100k fresh MCMC steps under the converged bias (≈30 min) due to slow mixing across modes.

### 4.3. Cu-Au Alloy

**Demonstration on realistic materials system.** Unlike the Ising and Potts models, which serve as pedagogical benchmarks with analytical energy functions, the copper-gold (Cu-Au) binary alloy represents a realistic materials system with a complex energy landscape. The energy function for Cu-Au alloy is evaluated using cluster expansion models (Chang et al., 2019; Ångqvist et al., 2019) fitted to first-principles density functional theory (DFT) calculations (Damewood et al., 2022), making each energy evaluation significantly more expensive than the simple pairwise interactions in Ising or Potts models. See Section E for a brief introduction. This computational cost, while orders of magnitude faster than direct DFT, is still substantial that minimizing the number of energy evaluations becomes critical for practical applications. For Cu-Au, we use the gold fraction $x_{Au}$ (fraction of sites occupied by Au) as the CV, which distinguishes the ordered phases (e.g., Cu$_3$Au at $x_{Au} \approx 0.25$, Figure 5(a), CuAu at $x_{Au} \approx 0.5$, Figure 5(b)) and the disordered phase.

MetaDNS successfully captures the thermodynamic behavior of Cu-Au across all temperatures and supercell sizes. As shown in Table 10, at the smaller $2 \times 2 \times 4$ supercell, both MDNS and MetaDNS perform well, with MDNS achieving slightly higher NESS values. However, at the larger

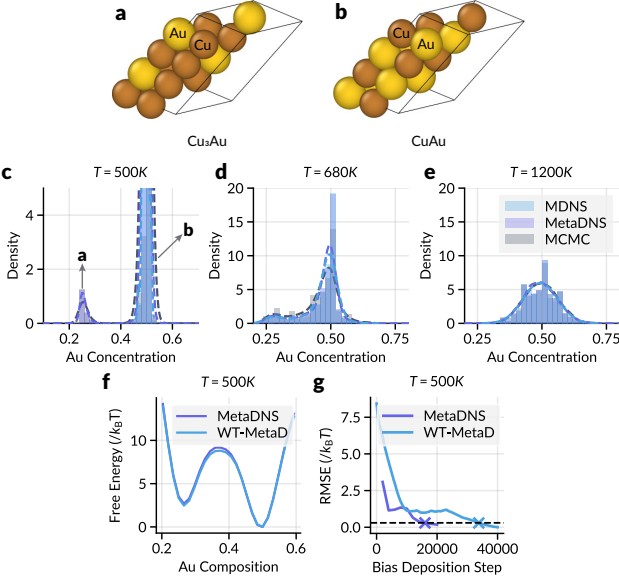

*Figure 5.* Cu-Au alloy results at $4 \times 4 \times 4$. (a-b) Crystal structures of $Cu_3Au$ and CuAu ordered phases. (c-e) Au concentration distributions at 500K, 680K, and 1200K showing MDNS mode collapse at 500K, capturing only the $x_{Au} = 0.5$ mode, whereas MetaDNS (ours) matches the ground truth. (f) Free energy profiles at 500K showing MetaDNS (ours) agreement with WT-MetaD. (g) Convergence speedup of MetaDNS over WT-MetaD at 500K (RMSE vs. steps).

$4 \times 4 \times 4$ supercell (see Figure 5, Table 11), MDNS exhibits mode collapse at low temperature (500K) by failing to capture the $Cu_3Au$ phase and only sampling the CuAu phase (Figure 5(c)), resulting in higher energy and $x_{Au}$ JS divergences (Table 11). In contrast, MetaDNS achieves substantially lower JS divergence ($7.9 \times 10^{-2}$ and $8.5 \times 10^{-2}$ for MetaDNS compared with $1.3 \times 10^{-1}$ and $1.3 \times 10^{-1}$ for MDNS), successfully sampling both ordered phases. In this case, $Cu_3Au$ is much less populated than the competing CuAu phase so missing it did not hurt the JS divergence values as much compared with the Ising and Potts cases. At higher temperatures (680K and 1200K), all methods show good agreement as the system transitions to a disordered phase where the modes coalesce.

The free energy profiles along $x_{Au}$ at 500K (Figure 5(f–g)) reveal that MetaDNS accurately reproduces the double-well potential with the minima corresponding $x_{Au} \approx 0.25$ ($Cu_3Au$) and 0.5 (CuAu) obtained by WT-MetaD. The convergence analysis (Figure 5(g)), measured with respect to the final WT-MetaD profile at 40k steps, demonstrates that MetaDNS reaches RMSE $< 0.3 \, k_B T$ in 16k bias deposition steps, compared to 33.8k steps for WT-MetaD, a $2.1\times$ reduction in bias deposition steps to convergence (see Table 13 and Section F.6). For Cu-Au, MetaDNS has an advantage versus WT-MetaD in wall-clock training time despite requiring neural network optimization (1.5 h vs. 1.75 h on an

A100 GPU). Additionally, MetaDNS provides a decisive inference advantage: generating 10k samples via 64 autoregressive steps (one step per lattice site) takes under 1 min. In contrast, WT-MetaD requires ~1k MCMC steps ($\approx 40$ min) since the cluster expansion must be called sequentially at every step, resulting in a $>40\times$ inference speedup in favor of MetaDNS that grows with the cost of energy evaluation.

## 5. Discussion

### 5.1. Key Contributions

MetaDNS integrates well-tempered metadynamics with discrete neural samplers, addressing a fundamental limitation of current state-of-the-art methods: mode collapse at low temperatures. By maintaining a history-dependent bias potential along low-dimensional collective variables, we provide an explicit, interpretable exploration signal that fills in visited regions and encourages barrier crossing, as opposed to current methods that are limited by convergence to a fixed target or by annealing paths. Our formulation transfers the classical metadynamics bias commonly used in continuous state spaces to discrete settings and shows how importance reweighting preserves asymptotically exact sampling from the target Boltzmann distribution. We empirically demonstrate the strength and usefulness of MetaDNS across multiple systems: on Ising and Potts models, MetaDNS recovers full bimodal or multi-modal distributions and enables free energy estimation where MDNS fails, while maintaining correct statistics after reweighting; on the Cu-Au binary alloy, MetaDNS captures both ordered phases at low temperature. Furthermore, by leveraging neural sampling to produce independent configurations rather than correlated MCMC chains at fixed WT-MetaD biases, MetaDNS achieves comparable or better exploration with drastically fewer bias deposition steps to convergence on Potts and Cu-Au. The wall-clock tradeoff depends on the cost of $E(x)$: for Potts, energies are cheap and batched, so WT-MetaD bias deposition converges much faster in wall time while MetaDNS pays for inner-loop neural optimization; for Cu-Au, cluster expansion evaluations are expensive and sequential, and MetaDNS training wall time is slightly shorter than WT-MetaD's despite the network overhead. In both settings, inference is amortized after training and $>30\times$ faster than fresh MCMC at the converged bias for Potts, and $>40\times$ for Cu-Au.

### 5.2. Limitations and Future Work

Several limitations suggest directions for future work. **Collective variable selection** currently relies on domain knowledge or simple heuristics (e.g., magnetization for Ising, occupation counts for Potts, gold fraction for Cu-Au); poor CV choices can hinder exploration or reduce accuracy. **Computational overhead** from bias updates and importance reweighting is modest in our experiments but may grow

with CV dimensionality and bin count. **Scaling to very large or more complex materials systems** (e.g., larger lattice sizes or higher-dimensional CVs) remains to be studied; we expect CV design and sampler capacity to become more critical in that regime. A promising direction is **automatic CV discovery**: learning or refining collective variables from data or from the sampler's visitation statistics could reduce the need for hand-crafted CVs and improve applicability to new domains. Applying MetaDNS to more realistic systems such as metal oxides or high-entropy alloys would require more expensive MLFFs in place of cluster expansion.

## Software and Data

Code is available at https://github.com/xiaochendu/metadns. Pre-trained model checkpoints, sample pickles, and reproduction notebooks are available on Zenodo at https://doi.org/10.5281/zenodo.20301979.

## Acknowledgements

X.D. acknowledges funding from Amazon as part of the MIT Climate and Sustainability Consortium (MCSC). J.N. acknowledges support from the Mathworks Fellowship. W.G. acknowledges Georgia Tech ARC-ACO Fellowship for the support. W.G. and Y.C. are grateful for partial supports by NSF Grants ECCS-1942523, DMS-2206576, 2450378, AFOSR Grant FA9550-25-1-0169. M.T. is partially supported by NSF Grant DMS-2513699, DOE Grants NA0004261, SC0026274, Richard Duke Fellowship, and Simons Institute for the Theory of Computing at UC Berkeley. The authors acknowledge the MIT SuperCloud and Lincoln Laboratory Supercomputing Center for providing HPC resources. This research used resources of the National Energy Research Scientific Computing Center, a DOE Office of Science User Facility supported by the Office of Science of the U.S. Department of Energy under Contract No. DE-AC02-05CH11231 using NERSC award ALCC-ERCAP0038200.

## Impact Statement

Better sampling of discrete configurational spaces (alloys, order-disorder systems) supports a more reliable understanding of phase behavior and free energy landscapes, which in turn can accelerate the discovery and design of materials for energy, catalysis, and structural applications. By improving mode coverage and reducing the cost of thermodynamic sampling, MetaDNS may contribute to such efforts. Future work on automatic CV discovery, scaling to larger systems, and connections to other exploration mechanisms may further extend the applicability of physics-inspired neural samplers.

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

# A. Well-Tempered Metadynamics and Convergence of MetaDNS

Here we provide additional background on WT-MetaD for the interested reader and address convergence concerns raised in Section 3 when using a neural sampler instead of MCMC or MD.

## A.1. Background: Well-Tempered Metadynamics

WT-MetaD builds a bias $V(s)$ so that sampling from the biased distribution

$$\pi_V(x) = \frac{1}{Z_V} \exp\big(-\beta(E(x) + V(\xi(x)))\big) \tag{1}$$

yields broader exploration over the CV space $\mathcal{S}$. The bias converges to a fixed point that flattens the free energy:

$$V^\star(s) = -\left(1 - \gamma^{-1}\right) F(s) + c, \tag{2}$$
$$F(s) = -\frac{1}{\beta} \log \sum_{x:\xi(x)=s} \exp(-\beta E(x)),$$

with $F(s)$ the free energy along the CV, $\gamma > 1$ the bias factor, and $c$ a constant. Thus $F(s) + V^\star(s) = F(s)/\gamma + c$ is flattened by $\gamma$, and $V^\star \propto F$ allows recovery of the free energy from the learned bias.

## A.2. Convergence with a Neural Sampler

Classical convergence results for metadynamics hold for ergodic dynamics (MCMC or MD) (Micheletti et al., 2004; Barducci et al., 2008; Crespo et al., 2010; Dama et al., 2014). With a neural sampler, two issues arise: (1) **Non-ergodicity**: the neural sampler need not guarantee the visitation of all states as in MCMC or MD; and (2) **Approximation error**: the sampler may not learn the biased target $E_{\text{biased}}(x) = E(x) + V_{t-1}(\xi(x))$ at each $t$, so we effectively sample from $q_\theta \approx \pi_{V_t}$ rather than exactly from $\pi_{V_t}$.

In practice, the neural sampler $q_\theta$ at iteration $t$ is only an approximation to $\pi_{V_t}$, which introduces "biased noise" into the bias update. This error can be mitigated by: (1) **Inner-loop optimization**: a larger number of ($N_{\text{inner}}$) training steps per outer iteration to bring $q_\theta$ closer to $\pi_{V_t}$ before updating the bias; (2) **Conservative hill deposition**: the well-tempered factor $\exp(-V_t(s)/(\gamma k_B T))$ reduces hill heights as bias accumulates, making updates more robust to sampling errors; and (3) **Bounded cumulative error**: under bounded $\|q_\theta - \pi_{V_t}\|$ and appropriate decay of the effective step size, $V_t$ converges to a neighborhood of $V^\star$ with controlled asymptotic error.

For stronger guarantees, Metropolis–Hastings (MH) correction can be applied to neural proposals at each outer step, ensuring exact sampling from $\pi_{V_t}$ (Nicoli et al., 2020), at the cost of additional energy evaluations.

**Three levels of theoretical guarantee.**

1. **Exact sampling (strongest):** With exact sampling from $\pi_{V_t}$ at each step (e.g., MH-corrected proposals), $V_t \to V^\star$ and importance-weighted estimates are asymptotically exact.
2. **Controlled approximation (practical):** If $q_\theta$ stays close to $\pi_{V_t}$ via sufficient inner training, $V_t$ converges to a neighborhood of $V^\star$ with bounded error.
3. **Empirical validation (this work):** We use importance reweighting (likelihood-based where tractable) and validate against ground truth (Swendsen–Wang, MCMC). This is simple, scalable, and performs well without MH correction.

## A.3. Outstanding Theoretical Questions

Several conditions merit further study: (1) **timescale separation**: the theory assumes the sampler adapts faster than the bias ($\tau_{\text{learning}} \ll \tau_{\text{bias update}}$). We use $N_{\text{inner}}$ steps per update, but formal requirements remain open. (2) **Approximation error bounds**: we assume $\|q_\theta - \pi_{V_t}\|$ stays bounded; NESS and related diagnostics can inform this in practice, and longer training may improve closeness to $\pi_{V_t}$. (3) **Ergodicity and mode coverage**: we focus on mode discovery and sampling of rare intermediate states, hence formal ergodicity of the neural sampler is not required for that goal. These are useful directions for future work but do not affect the empirical utility of MetaDNS demonstrated in this work.

## B. Path Likelihood and Radon–Nikodým Derivatives for Diffusion Samplers

For diffusion-based samplers like MDNS (Zhu et al., 2025) that operate on continuous-time paths, the sampling probability is defined over paths rather than configurations. In this work we use *masked diffusion*, where the model iteratively unmask sites to generate configurations. A path $X = (X_0, X_1, \ldots, X_T)$ represents the sequence of configurations evolving over time (e.g., from masked to fully revealed), where $X_T$ is the final configuration. MDNS derives importance weights using the RN derivative between path measures. The RN derivative between the optimal path measure $\mathbb{P}^*$ and the current path measure $\mathbb{P}^u$ is

$$\frac{d\mathbb{P}^*}{d\mathbb{P}^u}(X) = \exp(W^u(X) - \log Z),$$

where the logarithmic RN derivative is

$$\log \frac{d\mathbb{P}^*}{d\mathbb{P}^u}(X) = W^u(X) - \log Z,$$

with

$$W^u(X) = r(X_T) + \sum_{t:X_{t-} \neq X_t} \log \frac{1/N}{s_\theta(X_{t-})_{d(t),X_t^{d(t)}}}.$$

Here, $r(X_T)$ is a terminal reward (typically $r(X_T) = -\beta E(X_T)$ to encode the Boltzmann target distribution), $s_\theta(X_{t-})_{d(t),X_t^{d(t)}}$ is the model's learned transition probability (score) for the transition from $X_{t-}$ to $X_t$ at dimension $d(t)$, and $N$ is the number of possible transitions (the number of possible spins or atoms in the system in the case of masked diffusion). In MDNS implementations, $\log_{\mathrm{rnd}} = W^u(X)$. The RN derivative $\exp(W^u(X) - \log Z)$ can be interpreted as path-level importance weights, directly usable for ESS calculation and importance sampling over paths.

The RN derivative provides a reweighting factor: if a path $X$ is generated under measure $\mathbb{P}^u$ with some probability, then its relative probability under measure $\mathbb{P}^*$ is given by $(d\mathbb{P}^*/d\mathbb{P}^u)(X)$. The optimal measure $\mathbb{P}^*$ corresponds to paths that terminate at configurations distributed according to the Boltzmann distribution $\pi(x) \propto \exp(-\beta E(x))$.

To recover the configuration-level likelihood $q_\theta(x)$ for a final configuration $x = X_T$, we work directly with the RN derivative relationship. The key insight is that the sum over transitions in $W^u(X)$ relates to the autoregressive likelihood. Specifically, for autoregressive-style samplers, the sum $\sum_{t:X_{t-} \neq X_t} \log s_\theta(X_{t-})_{d(t),X_t^{d(t)}}$ corresponds to the log-probability of the path under the model's transition dynamics, which for autoregressive models equals $\log q_\theta(x) = \sum_i \log q_\theta(x_i|x_{<i})$ up to constant terms.

From the RN derivative relationship and the structure of $W^u(X)$, we can express the configuration-level log-likelihood as:

$$\log q_\theta(x) = -\beta E(x) - W^u(X) + \text{const} = -\beta E(x) - \log_{\mathrm{rnd}} + \text{const},$$

where $x = X_T$ is the final configuration of path $X$, and $\log_{\mathrm{rnd}} = W^u(X)$. The constant term arises from normalization and transition counting, and can be ignored for self-normalized importance sampling since only weight ratios matter. Thus the path-based RND framework from MDNS yields configuration-level $q_\theta(x)$ compatible with standard likelihood-based importance sampling where weights are $\tilde{w}_i = \exp(-\beta E(x_i))/q_\theta(x_i)$.

## C. Evaluation Metrics and Free Energy Terminology

The following metrics are used in Section 4. All expectations and divergences involving MetaDNS are computed on importance-weighted samples unless stated otherwise.

**Normalized effective sample size (NESS).** For weighted samples $\{(x_i, w_i)\}_{i=1}^N$, the effective sample size is $\text{ESS} = \left(\sum_{i=1}^N w_i\right)^2 / \sum_{i=1}^N w_i^2$. We report $\text{NESS} = \text{ESS}/N \in [0, 1]$. Higher NESS indicates more efficient use of samples but can also be indicative of mode collapse; low NESS can indicate high weight variance (e.g., when reweighting from a biased to the target distribution).

**Jensen–Shannon divergence (JS Div.).** We report the Jensen–Shannon divergence between the empirical distribution of the (reweighted) sampler and the ground-truth distribution (SW for Ising/Potts, MCMC for Cu-Au). We report it for the energy distribution (E. JS Div.), over the distribution of CV(s) in the samples ($x_\uparrow$ or $x_{\text{Au}}$ JS Div., CV JS Div.), etc. Lower values indicate better agreement with the target.

**Magnetization (Mag.) and two-point correlation (Corr.).** For Ising and Potts models, Mag. is the absolute magnetization per site ($|m|$ with $m$ the mean spin). Corr. is the average two-point correlation over the lattice, evaluated at increasing distances (not only nearest-neighbor). **Ising:** spins $\in \{-1, +1\}$; Corr. is the lattice average of the spin product $s_i s_j$ over pairs at each distance, with periodic boundaries. **Potts:** states $\in \{0, \dots, q-1\}$; Corr. at each distance is the average over the four (horizontal and vertical) neighbors of the indicator that the site and neighbor match, subtracting $1/q$, so uncorrelated configurations yield zero; we average over distances. Both Mag. and Corr. are computed as expectations under the (reweighted) sampler and compared to the ground truth.

**Potential of mean force (PMF) along the CV.** In Figures 4 and 5, "free energy" profiles and plots refer to the *PMF* along the chosen collective variable $\xi(x)$, not the full thermodynamic free energy of the ensemble. For a discrete CV, $s$, the PMF is $F(s) = -\beta^{-1} \log \sum_{x:\xi(x)=s} \exp(-\beta E(x))$ (up to an additive constant). This is the same $F(s)$ as in the WT-MetaD fixed point (Equation (2)). Reconstructing $F(s)$ from the learned bias or from reweighted samples is the free energy *landscape* along the CV, which is used for barrier heights and phase identification.

**RMSE for PMF convergence (Figures 4 and 5).** Convergence panels plot RMSE between the running PMF estimate and a fixed reference curve. The reference is the WT-MetaD PMF at a large step count (125k for Potts; 40k for Cu-Au at 500K). Because the PMF is defined only up to an additive constant, we vertically align MetaDNS and the reference by shifting each curve so that their minima coincide, then compute RMSE over the discrete CV grid points,

$$\text{RMSE} = \sqrt{\frac{1}{|\mathcal{G}|} \sum_{s \in \mathcal{G}} \big(F(s) - F_{\text{ref}}(s)\big)^2},$$

reported in units of $k_{\text{B}}T$. Here $\mathcal{G}$ is the set of bins used for the PMF in CV space, so RMSE is a per-grid-point average discrepancy. For Potts, RMSE is computed on the full 2D CV discretization (the same grid used for the bias) while the 1D slice along CV 1 in the figure is for visualization only.

## D. Collective Variable Definitions

**Potts model** ($q = 3$). The 2D collective variable (CV 1, CV 2) used in the Potts experiments is a projection of the state concentrations onto the plane. Let $c_1$, $c_2$, $c_3$ denote the fractions of lattice sites in Potts states 0, 1, and 2 respectively ($c_1 + c_2 + c_3 = 1$). We define

$$\text{CV 1} = c_1 - \tfrac{1}{2}(c_2 + c_3), \qquad \text{CV 2} = \tfrac{\sqrt{3}}{2}(c_2 - c_3).$$

This is the standard projection for the 3-simplex: the three ordered phases (all sites in one state) map to the vertices of an equilateral triangle, and the disordered phase $(c_1, c_2, c_3) = (1/3, 1/3, 1/3)$ maps to the origin $(0, 0)$. The bias potential and free energy profiles are defined over a discretization of this 2D CV space.

## E. Cluster Expansion and Formation Energy (Cu-Au)

In the Cu-Au experiments, the energy $E(x)$ is the *formation energy* of configuration $x$: the energy of that atomic arrangement relative to the pure-component reference states. It is computed via a *cluster expansion* (CE) (Chang et al., 2019; Ångqvist et al., 2019), a surrogate model that renders sampling over the large configurational space of the alloy tractable.

First-principles methods such as density functional theory (DFT) yield accurate energies but are too costly to apply to every configuration in a sampling run. The configurational space of a binary alloy on a fixed lattice grows exponentially with the number of sites (e.g., $2^N$ for $N$ sites). Cluster expansion addresses this by fitting a fast Hamiltonian to a limited set of DFT reference energies; once fitted, $E(x)$ for any configuration $x$ can be evaluated in milliseconds rather than minutes or hours.

Formally, the lattice is a graph whose sites carry discrete labels (e.g., Cu or Au). The CE expresses the formation energy as a sum over small subgraphs ("clusters"): single-site terms (chemical potentials), pairs, triplets, and optionally higher-order clusters. Each cluster type $\alpha$ has a learned weight $J_\alpha$ (effective cluster interaction, ECI), so $E(x) = \sum_\alpha J_\alpha \phi_\alpha(x)$, where $\phi_\alpha(x)$ is a basis function that aggregates over all clusters of type $\alpha$ in the lattice (e.g., sum or average of occupancy products over pairs of that type); in a binary system each site contributes a factor such as $\pm 1$, so $\phi_\alpha(x)$ takes values that depend on the configuration and the number of such clusters. Training consists of supervised regression of CE energies onto DFT energies on a training set of configurations, often with regularization (e.g., Lasso) for sparsity (Chang et al., 2019; Ångqvist et al., 2019). The fitted model generalizes to the full discrete configurational space and respects locality and lattice symmetry.

In this work we use a CE fitted to DFT for Cu-Au (Damewood et al., 2022). Every MetaDNS or MCMC step that evaluates $E(x)$ uses this CE rather than DFT. The formation energies define the Boltzmann distribution $\pi(x) \propto e^{-\beta E(x)}$; configurations with lower formation energy are thermodynamically favored, so sampling concentrates on stable ordered or disordered phases. Reducing the number of $E(x)$ evaluations (e.g., via MetaDNS) therefore reduces the computational cost of exploring the alloy phase space.

# F. Implementation Details

### F.1. CV-stratified replay buffer

For the larger Ising (16×16 at low temperature) and Potts (8×8 and 16×16) systems, we optionally use a CV-stratified replay buffer alongside metadynamics to improve training. The buffer stores past samples; a fraction of each training batch (the *buffer ratio*) is drawn from the buffer so the model receives gradient signal from under-explored CV regions. The buffer is partitioned into CV bins and sampled using a *balanced* strategy so that rare CV values are equally represented. Buffer size is 1024 with FIFO eviction. Both Ising and Potts use the same buffer ratio (0.5) and number of CV bins (8); since Ising uses a 1D CV (magnetization), this gives 8 bins total, whereas Potts uses a 2D CV (concentration projection), giving 8 bins per dimension. For **Ising**, we use this buffer only for the **low-temperature** 16×16 MetaDNS case; other Ising settings use no buffer.

### F.2. Hyperparameters and training

Key hyperparameters are summarized in Tables 2 to 4. Ising and Potts use a Vision Transformer with RoPE (RopeVIT): patch size 1, embed dim 64 (Ising) or 128 (Potts), depth 4, heads 4 and random-order autoregressive sampling. Cu-Au uses a Periodic 3D RoPE Transformer over a 3D grid (see Section F.2.1); hidden dim 64, depth 4, heads 4. Training: WDCE loss; $N_{\text{inner}}$ (training steps per bias update) and $N_{\text{outer}}$ (number of outer-loop bias updates) as in the tables; 8 WDCE replicates; Adam, lr $1 \times 10^{-4}$, no weight decay, EMA decay 0.9999. Metadynamics uses hill width $\sigma = 0.05$, initial hill height $h = 0.1\,k_B T$, and bias factor $\gamma = 10$ in all cases. For the MDNS baseline we use the same hyperparameters but without the metadynamics bias grid.

*Table 2.* Key hyperparameters: Ising. Buffer used only for low-$T$ 16×16 MetaDNS (ratio 0.5, 8 bins, 1D CV).

|  | 4×4 | 8×8 | 16×16 |
| --- | --- | --- | --- |
| **Model** | | | |
| $N$, embed, depth, heads | 16, 64, 4, 4 | 64, 64, 4, 4 | 256, 64, 4, 4 |
| **Training** | | | |
| Batch, buffer size | 128, 0 | 128, 0 | 128, 1024 |
| Buffer ratio, CV bins | – | – | 0.5, 8 |
| WDCE repl., $N_{\text{inner}}$ | 8, 5 | 8, 5 | 8, 5 |
| $N_{\text{outer}}$ | 20k | 20k | 50k |

*Table 3.* Key hyperparameters: Potts ($q = 3$). MetaDNS uses buffer for 8×8 and 16×16 (ratio 0.5, 8 bins/dim, 2D CV).

|  | 4×4 | 8×8 | 16×16 |
| --- | --- | --- | --- |
| **Model** | | | |
| $N$, embed, depth, heads | 16, 128, 4, 4 | 64, 128, 4, 4 | 256, 128, 4, 4 |
| **Training** | | | |
| Batch, buffer size | 128, 0 | 128, 1024 | 128, 1024 |
| Buffer ratio, CV bins | – | 0.5, 8 | 0.5, 8 |
| WDCE repl., $N_{\text{inner}}$ | 8, 5 | 8, 5 | 8, 5 |
| $N_{\text{outer}}$ | 20k | 20k | 50k |

*Table 4.* Key hyperparameters: Cu-Au. CV: 1D composition $x_{Au}$.

|  | 2×2×4 | 4×4×4 |
| --- | --- | --- |
| **Model** | | |
| $N$, hidden, depth, heads | 16, 64, 4, 4 | 64, 64, 4, 4 |
| **Training** | | |
| Batch, buffer | 128, 0 | 128, 0 |
| WDCE repl., $N_{inner}$ | 8, 5 | 8, 5 |
| $N_{outer}$ | 20k | 20k |

### F.2.1. PERIODIC POSITIONAL EMBEDDINGS FOR TRANSFORMERS

To respect the periodic boundary conditions inherent to lattice systems, we design sinusoidal positional embeddings with frequencies that satisfy periodicity constraints. For a lattice with period $L$ along a given dimension, we require

$$\sin(\omega_i x) = \sin(\omega_i (x + L)),$$
$$\cos(\omega_i x) = \cos(\omega_i (x + L))$$

for all positions $x$. This constraint is satisfied when $L\omega_i = 2n_i\pi$ for $n_i \in \mathbb{Z}$, yielding quantized frequencies

$$\omega_i = \frac{2n_i\pi}{L}, \quad n_i = 1, 2, 3, \ldots$$

We apply this construction independently to each lattice dimension (e.g., $x$, $y$, $z$ for 3D systems), ensuring that the positional embeddings respect the full periodic lattice topology. The resulting embeddings are incorporated into the Transformer architecture via Rotary Position Embedding (RoPE) (Su et al., 2023), maintaining equivariance under lattice translations.

### F.3. Baseline and ground-truth procedures

We use Swendsen–Wang (SW) (Swendsen & Wang, 1987) as ground truth for Ising and Potts, and MCMC-based well-tempered metadynamics (WT-MetaD) (Barducci et al., 2008) for convergence comparison and for Cu-Au reference. Key settings are summarized below.

**Ising.** SW: batch 1024, 32 blocks × 128 steps, burn-in 1024; $L \in \{4, 8, 16\}$, $\beta \in \{0.28, 0.4407, 0.6\}$.

**Potts ($q = 3$).** SW: batch 1024, 40 blocks × (250 or 500) steps, burn-in 2048; $\beta \in \{0.5, 1.005, 1.2\}$. WT-MetaD: batch 128, 125k deposition steps (16×16); 2D CV (concentration projection); CV grid 17×17 / 65×65 / 257×257 (for 4×4 / 8×8 / 16×16); $\sigma = 0.05$, $h \in [0.1, 0.5] \, k_B T$ (temperature-dependent), $\gamma = 10$; update every 64 MCMC steps.

**Cu-Au.** Unbiased MCMC: batch 1000, 500 steps/block, 3–6 blocks/temperature; $T \in [200, 1200]$ K. WT-MetaD: 40k deposition steps, batch 128; 1D CV (composition); CV grid 65; $\sigma = 0.05$, $h \in [0.1, 0.5] \, k_B T$ (temperature-dependent); $\gamma = 10$; update every 64 MCMC steps.

### F.4. Warm-Start Baselines

An alternative to MetaDNS is to pre-train the neural sampler on a softened, higher-temperature target $\pi_{\beta'}(x) \propto e^{-\beta' E(x)}$ with $\beta' < \beta$, and then fine-tune at the true $\beta$. This strategy is also described in Zhu et al. (2025) (Appendix D.2.4). While pre-warming reduces energy barriers globally during early training, it lacks a *history-dependent* mechanism to discourage revisiting already-explored modes. When fine-tuned at the target $\beta$, the model may re-collapse to whichever mode was dominant at the end of warm-up, since no bias persists to redirect it. MetaDNS's CV-space bias, by contrast, accumulates specifically in *visited* regions and exerts a directed pressure toward unexplored parts of configuration space independently of the global energy scale. Empirical evidence for warm-start MDNS across all three systems is reported in Table 1 (Ising $L = 16$), Table 9 (Potts $L = 16$; see also Figures 12 and 15c for sample-level mode collapse), and Table 11 (Cu-Au $4 \times 4 \times 4$): warm-start reduces but does not eliminate mode collapse in the Ising and Potts settings, and improves results slightly for Cu-Au, but MetaDNS consistently outperforms warm-start MDNS.

### F.5. MDNS Training Pipeline

MDNS trains an autoregressive masked diffusion sampler using the Weighted Denoising Cross-Entropy (WDCE) loss, which is grounded in stochastic optimal control of continuous-time Markov chains (Zhu et al., 2025). The core training loop alternates between generating masked trajectories from the current model and updating the score network to minimize an importance-weighted denoising objective. Algorithm 2 gives a condensed version of the MDNS training procedure; full details including the path-level RN derivative weights $e^{W^{\tilde{u}}(X)}$ are in Section B.

---

**Algorithm 2** MDNS Training (condensed from Zhu et al. (2025))

---

1: **Input:** Energy function $E(x)$, inverse temperature $\beta$
2: **Initialize:** Score network parameters $\theta_0$, replay buffer $\mathcal{B} \leftarrow \emptyset$
3: **for** $k = 1$ to $K$ **do**
4:     **if** replay buffer refresh step **then**
5:         Sample $B$ unmasking trajectories $\{X^{(b)}\}_{b=1}^B$ from current model $q_\theta$
6:         Compute path-level importance weights $w^{(b)} \propto e^{W^{\tilde{u}}(X^{(b)})}$
7:         Store final clean samples $\{X_T^{(b)}, w^{(b)}\}$ in $\mathcal{B}$
8:     **end if**
9:     Resample masked versions $\tilde{x}^{(b)}$ of stored samples from $\mathcal{B}$ at random mask times $\lambda$
10:    Compute WDCE loss: $\mathcal{L}_{\text{WDCE}}(\theta) = -\frac{1}{Z} \sum_b w^{(b)} \cdot \mathbb{E}_\lambda \left[ \sum_d \log s_\theta(\tilde{x}^{(b)})_{d, X_T^{(b),d}} \right]$
11:    Update $\theta \leftarrow \theta - \nabla_\theta \mathcal{L}_{\text{WDCE}}(\theta)$
12: **end for**
13: **Output:** Trained sampler $q_\theta$ with tractable path likelihood

---

Setting $V_t \equiv 0$ in Algorithm 1 (i.e., training on the unbiased energy $E(x)$ throughout) recovers this MDNS training pipeline exactly, with the WDCE loss serving as the inner-loop objective $\mathcal{L}(\theta; \{x_i\}, E_{\text{biased}})$.

### F.6. Computational resources

All experiments were run on A100 GPUs. Potts energy evaluations are GPU-accelerated and fully batched; Cu-Au cluster expansion evaluations are sequential and CPU-bound. Tables 12 and 13 (Additional Benchmark Tables) report wall-clock training and inference times comparing MetaDNS and MCMC-based WT-MetaD. MetaDNS training steps are outer-loop iterations, each comprising $N_{\text{inner}}$ sampler training mini-batches plus one bias deposition update; WT-MetaD training steps are sequential MCMC steps (batched for Potts, sequential for Cu-Au). MetaDNS inference steps are autoregressive unmasking passes (equal to the number of lattice sites). Using bias-based reweighting $w_i = \exp(V(\xi(x_i)))$, no energy evaluations are needed at inference. WT-MetaD inference steps are MCMC sweeps under the converged static bias, each requiring an energy evaluation per chain.

**Cost accounting.** The RMSE convergence panels in Figure 4(d–f) and Figure 5(g) plot RMSE against *bias deposition steps* (outer-loop cost), not raw energy evaluation steps. Each MetaDNS outer-loop step contains $N_{\text{inner}}$ inner steps of (sample from $q_\theta \rightarrow$ evaluate biased energy $E(x) + V_{t-1}(\xi(x)) \rightarrow$ compute WDCE loss and backpropagate) followed by one outer-loop hill deposition batch. For all systems, $N_{\text{inner}} = 5$ (see Tables 2 to 4). Each WT-MetaD bias deposition step comprises 64 single-flip Metropolis MCMC steps batched across 128 chains, i.e., $64 \times 128 = 8{,}192$ energy evaluations per deposition step (see Section F.3). By contrast, each MetaDNS deposition step uses $N_{\text{inner}} \times M_{\text{inner}} = 5 \times 128 = 640$ energy evaluations. The apparent per-step advantage of MetaDNS in RMSE plots therefore translates to a $\sim 13\times$ reduction in raw energy evaluations per bias deposition step, on top of any step-count advantage. Network forward/backward passes are $\mathcal{O}(N)$ in lattice size and dominate Potts wall-time (cheap energy, expensive optimization), but not Cu-Au wall-time (expensive sequential cluster expansion, cheaper optimization relative to energy cost).

## G. Additional Results

### G.1. Additional Ising Model Figures

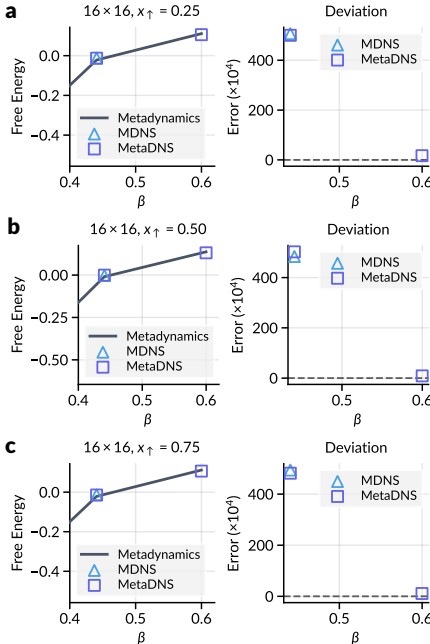

*Figure 6.* $L = 16$ Ising model free energy per site and deviation from WT-MetaD reference for $x_\uparrow =$ (a) 0.25, (b) 0.5, and (c) 0.75. MetaDNS (ours) is able to obtain samples across the full composition range at low $\beta = 0.6$ and critical $\beta = 0.4407$ temperatures for free energy (mixing energy) estimation but MDNS fails at low temperature.

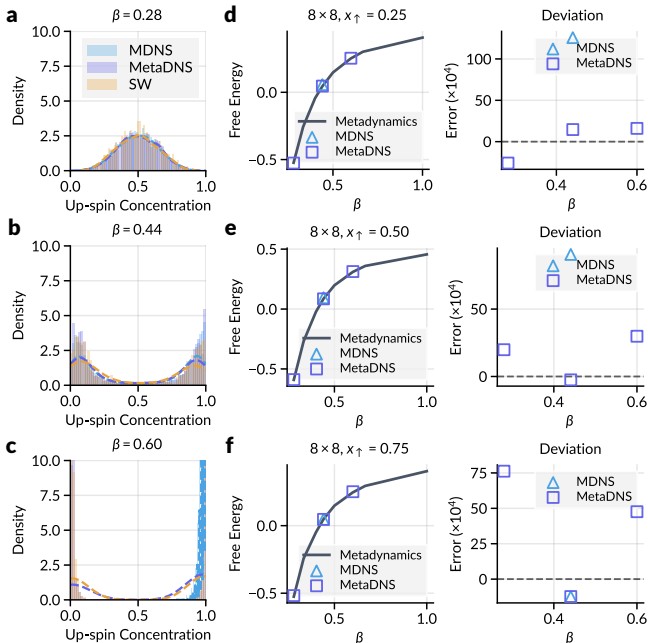

*Figure 7.* Ising model results at $L = 8$. (a-c) Up-spin concentration distributions across three temperatures, showing excellent agreement between MetaDNS (ours) and SW ground truth across all temperatures after reweighting but MDNS fails to sample the full down-spin mode at low temperature. (d-f) Free energy per site for $x_\uparrow = 0.25$, $0.5$, and $0.75$ compared with WT-MetaD reference and deviation from WT-MetaD reference. MetaDNS (ours) is able to obtain samples across the full composition range at low $\beta = 0.6$, critical $\beta = 0.4407$, and high $\beta = 0.28$ temperatures for free energy (mixing energy) estimation but MDNS can only do so at the critical temperature.

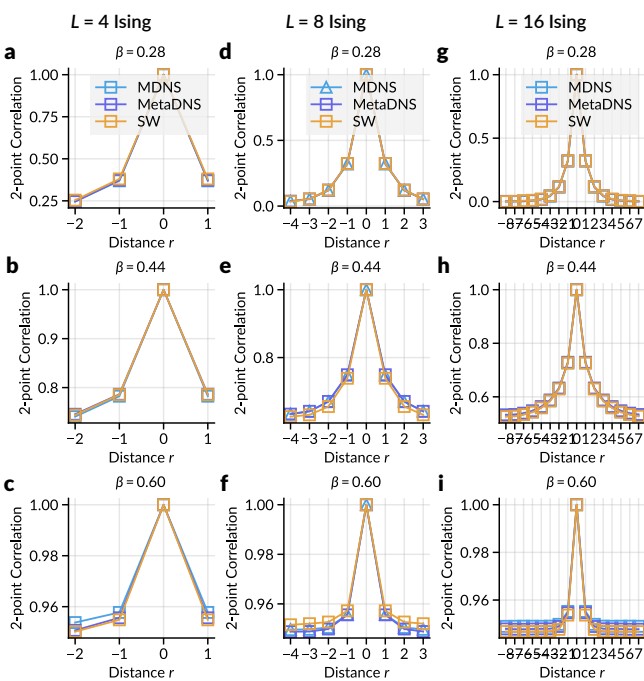

*Figure 8.* Two-point correlation functions for Ising models at $L \in \{$(a-c) 4, (d-f) 8, and (g-i) 16$\}$ and three temperatures. MetaDNS (ours, after reweighting) and MDNS both show strong agreement with SW ground truth across all conditions, validating that MetaDNS (ours) maintains correct statistical properties while achieving improved mode exploration.

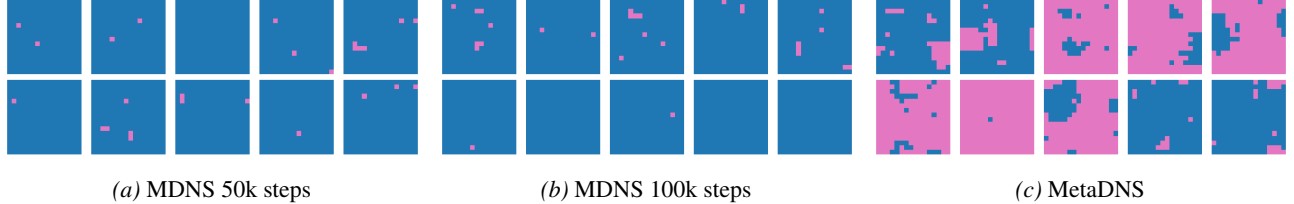

*(a)* MDNS 50k steps          *(b)* MDNS 100k steps          *(c)* MetaDNS

*Figure 9.* Visual comparison of $L = 16$ Ising model samples at low temperature. (a) MDNS shows mode collapse when trained for 50k steps, sampling predominantly from a single spin configuration. (b) Mode collapses persists even when MDNS was trained for 100k steps (2x original). (c) MetaDNS (ours) successfully discovers diverse configurations with well-formed domains of both spin orientations, demonstrating effective mode exploration.

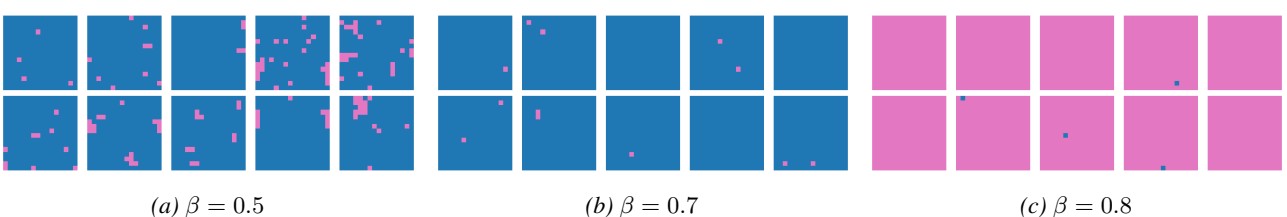

*(a)* $\beta = 0.5$          *(b)* $\beta = 0.7$          *(c)* $\beta = 0.8$

*Figure 10.* Visual comparison of $L = 16$ Ising model samples at additional low temperature $\beta > \beta_{\mathrm{crit}}$. (a) $\beta = 0.5$, (b) $\beta = 0.7$, and (c) $\beta = 0.8$. Regardless of exact temperature value, MDNS shows mode collapse.

## G.2. Additional Potts Model Figures

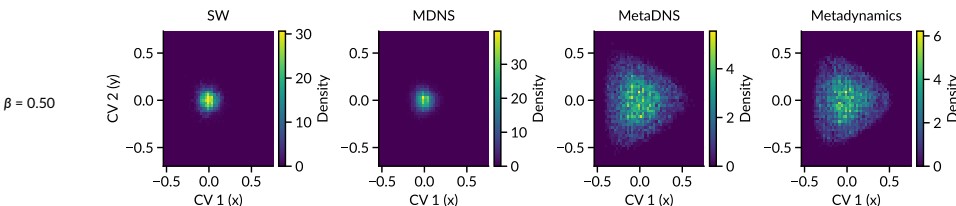

*Figure 11.* CV distributions for Potts ($q = 3$) models at $L = 16$ and high temperature. MDNS agrees with SW ground truth for this temperature, favoring the random configurations with CV $\approx (0, 0)$. MetaDNS (ours) and WT-MetaD are able to sample the CV landscape more broadly.

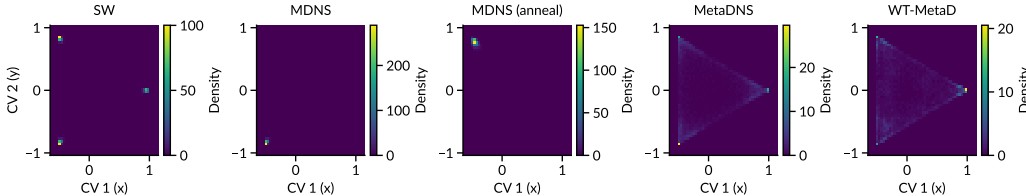

*Figure 12.* Mode collapse comparison for MDNS $L = 16$ Potts model (center) at $\beta = 1.2$ even when warm-started at high temperature.

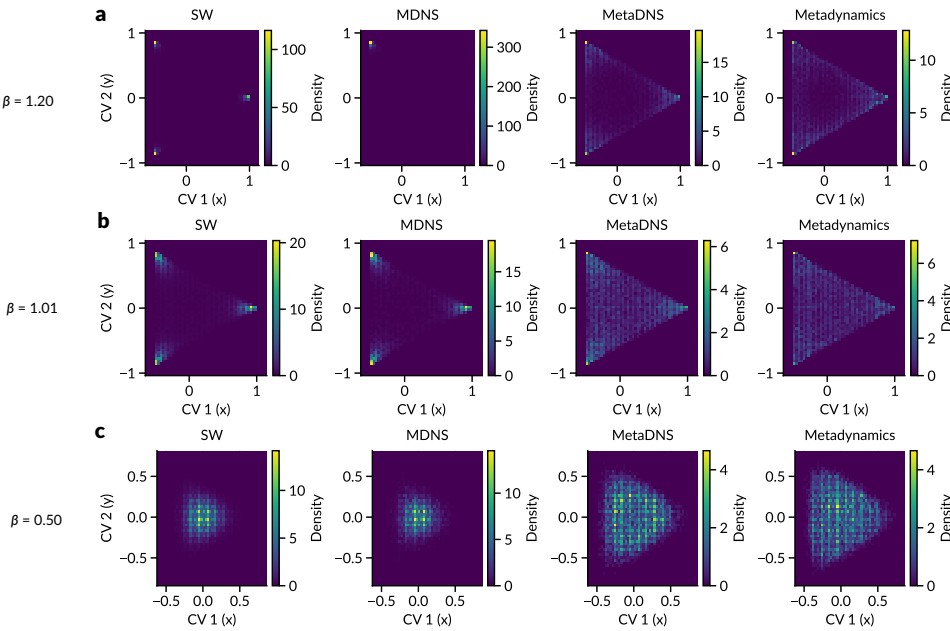

*Figure 13.* CV distributions for Potts ($q = 3$) models at $L = 8$ and three temperatures: (a) low $\beta = 1.2$, (b) critical $\beta = 1.005$, and (c) high $\beta = 0.5$. Similar to the $L = 16$ case, MetaDNS (ours) and WT-MetaD are able to sample the CV landscape more broadly at all temperatures, with all modes present. MDNS, however, exhibits mode collapse at low temperature $\beta = 1.2$ and only samples one of the three modes present in SW ground truth.

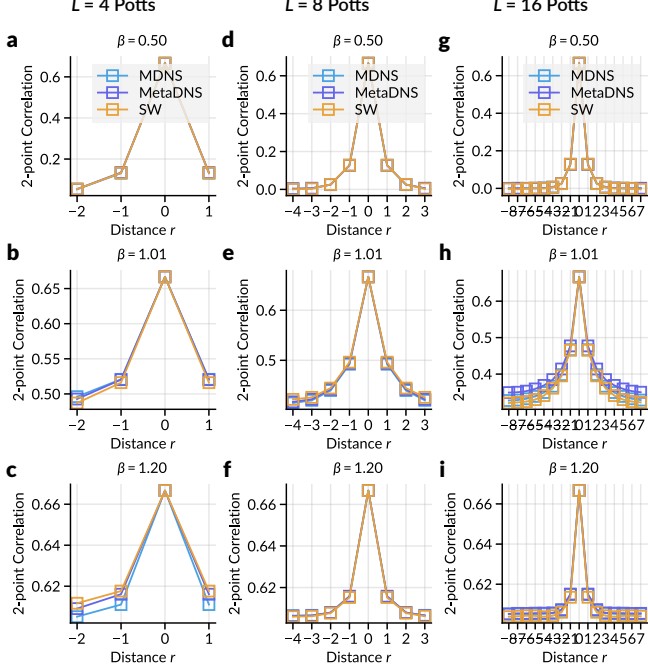

*Figure 14.* Two-point correlation functions for Potts ($q = 3$) models at $L \in \{$(a-c) 4, (d-f) 8, and (g-i) 16$\}$ and three temperatures. MetaDNS (ours, after reweighting) and MDNS both show strong agreement with SW ground truth across all conditions, validating that MetaDNS (ours) maintains correct statistical properties while achieving improved mode exploration.

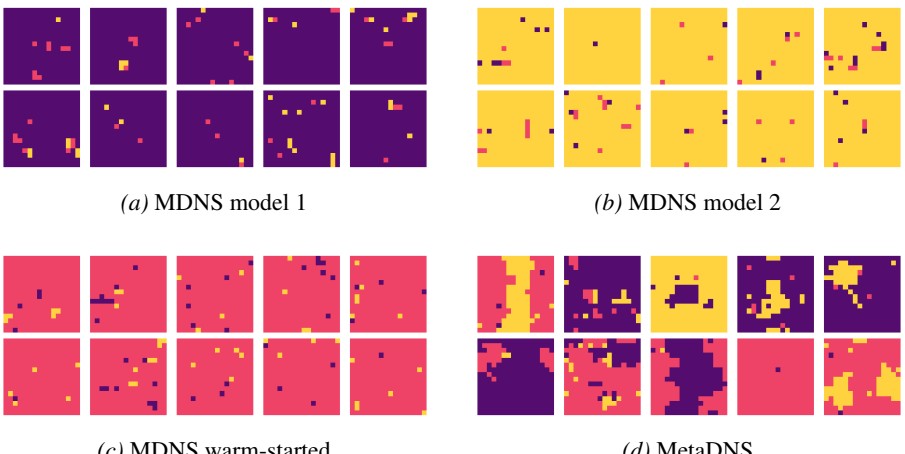

*(a)* MDNS model 1          *(b)* MDNS model 2

*(c)* MDNS warm-started        *(d)* MetaDNS

*Figure 15.* Visual comparison of $L = 16$ Potts ($q = 3$) model samples at low temperature. (a) MDNS model 1 shows mode collapse, sampling predominantly from a single phase configuration; (b) MDNS model 2 exhibits similar mode collapse in a different phase; (c) MDNS still suffers from mode collapse after warm-starting from high temperature; (d) MetaDNS (ours) successfully discovers diverse configurations with well-formed domains of multiple phases, demonstrating effective mode exploration across all three Potts states.

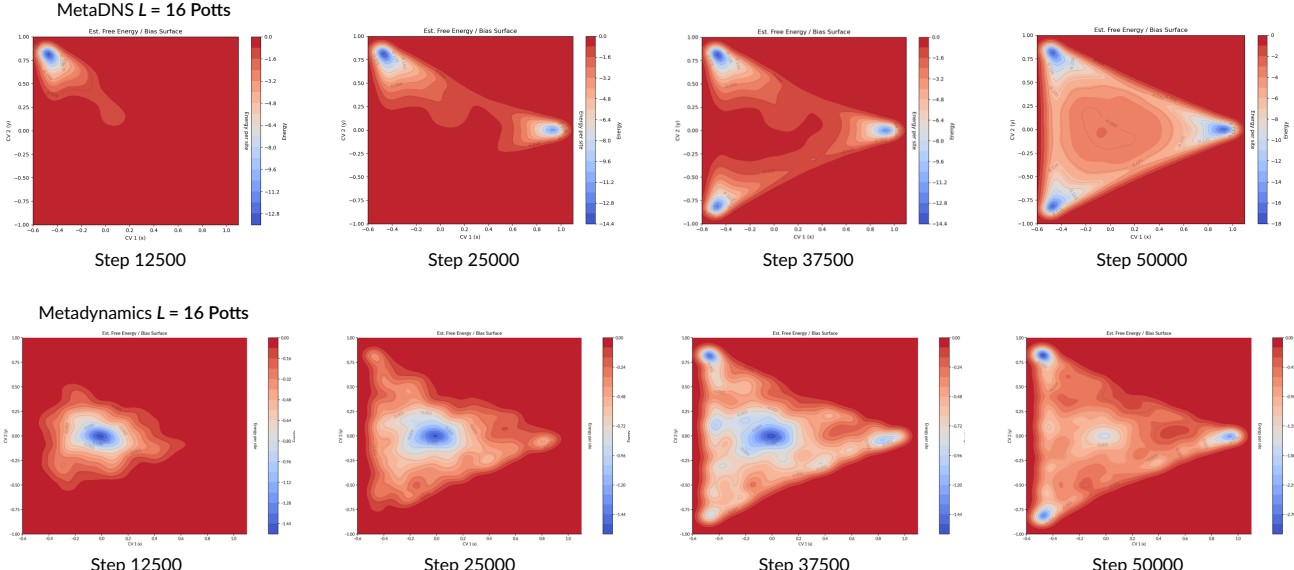

*Figure 16.* Learning dynamics for MetaDNS vs. MCMC-based WT-MetaD for the first 50k steps. WT-MetaD remains concentrated near random configurations (CV $\approx (0, 0)$) before gradually spreading to the target modes, whereas MetaDNS rapidly discovers and resolves the distinct low-energy basins.

## G.3. Sensitivity Analysis

We swept four WT-MetaD hyperparameters: hill width $\sigma \in \{0.01, 0.03, 0.05\}$, initial hill height $h \in \{0.1\,k_{\mathrm{B}}T,\ 0.5\,k_{\mathrm{B}}T\}$, bias factor $\gamma \in \{5, 10\}$, and, for Ising only, the number of CV bins $\in \{129, 257\}$. Across this sweep, NESS ranges are 0.30–0.70 for $16 \times 16$ Ising (consistent with the main-table MetaDNS value at $\beta = 0.6$), 0.20–0.50 for $16 \times 16$ Potts, and 0.20–0.40 for $4 \times 4 \times 4$ Cu-Au; all three ranges bracket the values reported in Tables 1, 9 and 11. Mode collapse is observed only at the extreme corner ($\sigma = 0.01$, $h = 0.1\,k_{\mathrm{B}}T$, $\gamma = 5$) for Potts (Figures 17 to 21); this corner also produces artificially high NESS, a known false positive under mode collapse where the effective support is narrow. Ising and Cu-Au shows no mode collapse across the full sweep (Figures 22 and 23).

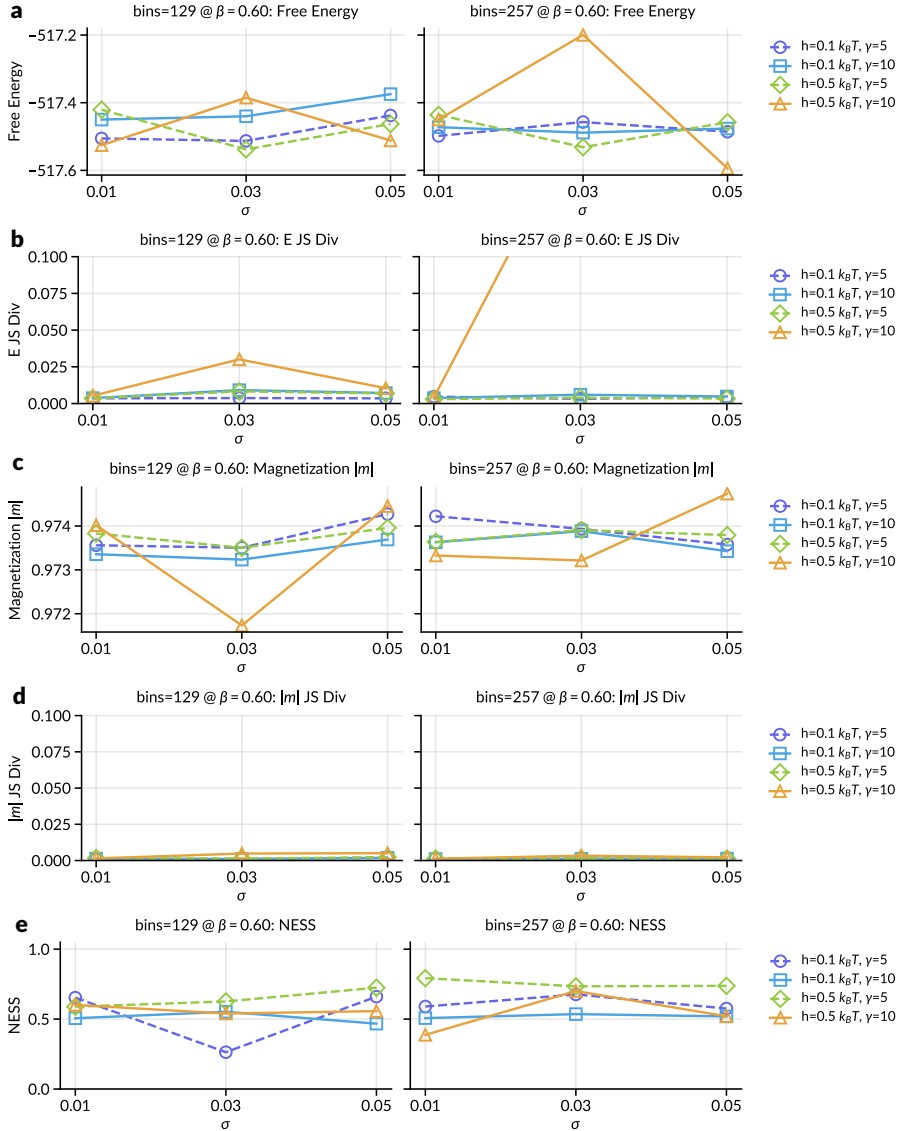

*Figure 17.* Sensitivity of MetaDNS to hyperparameters for the $L = 16$ Ising model at $\beta = 0.6$. $\sigma \in \{0.01, 0.03, 0.05\}$, $h \in \{0.1k_{\mathrm{B}}T, 0.5k_{\mathrm{B}}T\}$, $\gamma \in \{5, 10\}$, and number of CV bins $\in \{129, 257\}$.

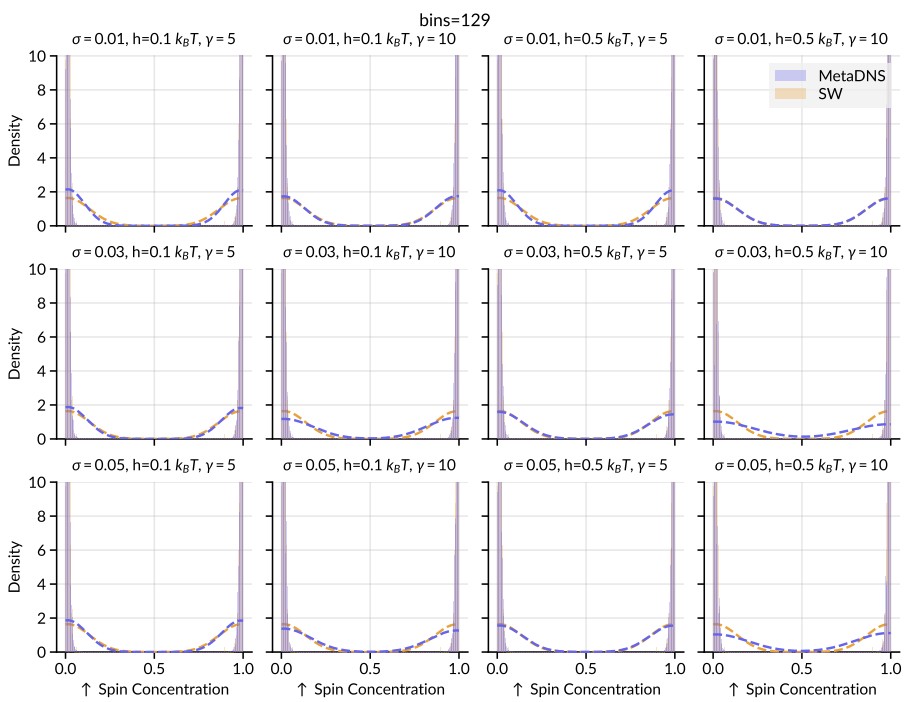

*Figure 18.* Up-spin concentration distributions sensitivity analysis for the $L = 16$ Ising model at $\beta = 0.6$ with 129 bins.

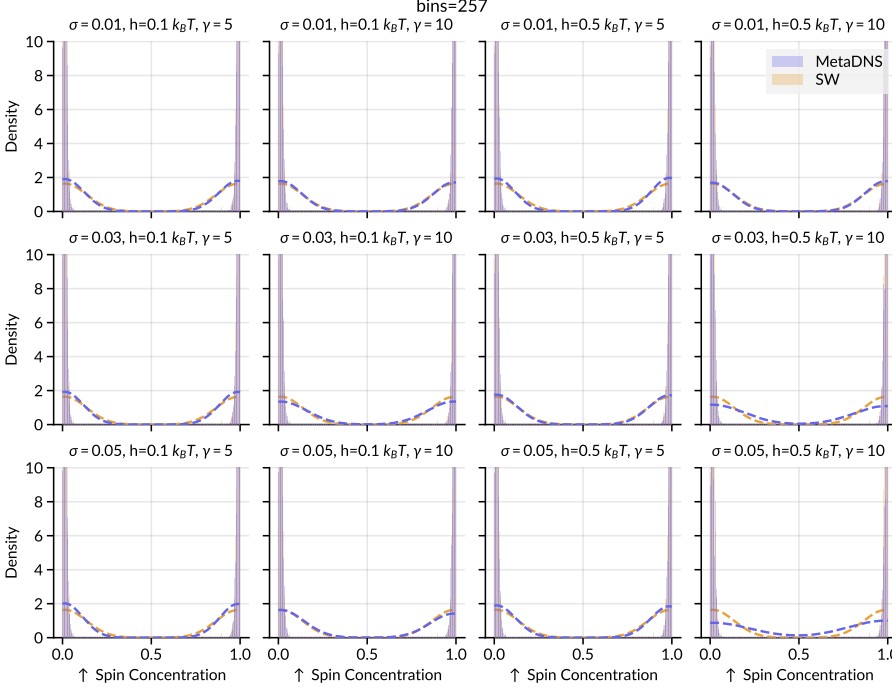

*Figure 19.* Up-spin concentration distributions sensitivity analysis for the $L = 16$ Ising model at $\beta = 0.6$ with 257 bins.

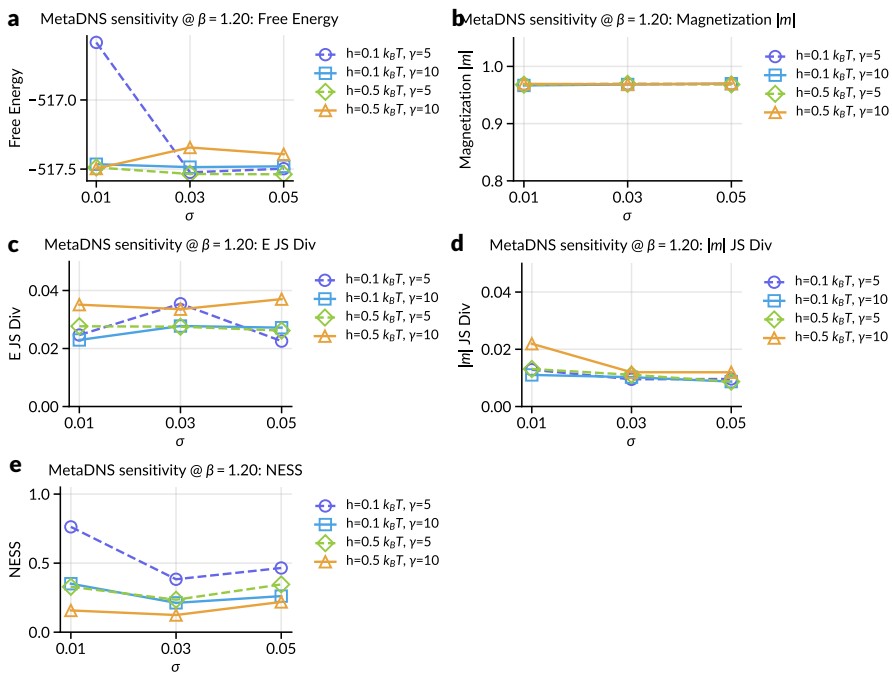

*Figure 20.* Sensitivity of MetaDNS to hyperparameters for the $L = 16$ Potts model at $\beta = 1.2$. $\sigma \in \{0.01, 0.03, 0.05\}$, $h \in \{0.1k_\mathrm{B}T, 0.5k_\mathrm{B}T\}$, $\gamma \in \{5, 10\}$.

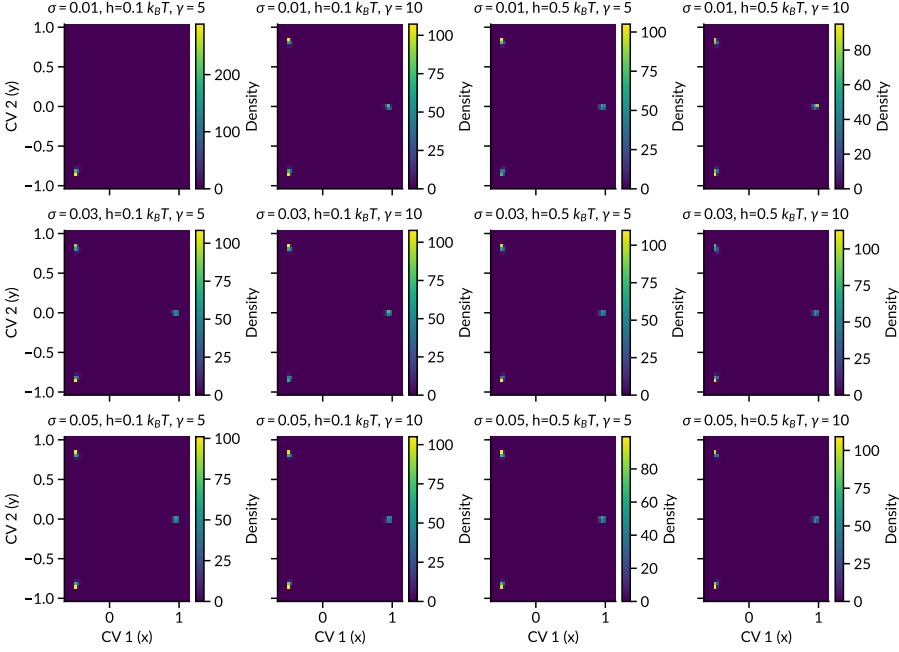

*Figure 21.* CV distributions sensitivity analysis for Potts ($q = 3$) models at $L = 16$ and low temperature. Mode collapse is observed only for the $\sigma = 0.01, h = 0.1k_\mathrm{B}T, \gamma = 5$ case.

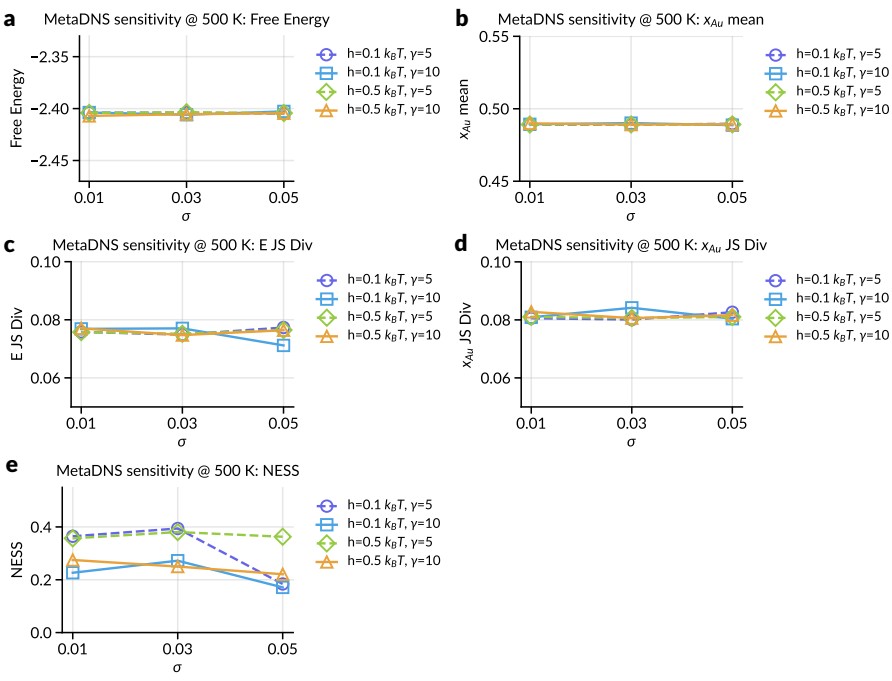

*Figure 22.* Sensitivity of MetaDNS to hyperparameters for the $4 \times 4 \times 4$ Cu-Au alloy at 500K. $\sigma \in \{0.01, 0.03, 0.05\}$, $h \in \{0.1k_\mathrm{B}T, 0.5k_\mathrm{B}T\}$, $\gamma \in \{5, 10\}$.

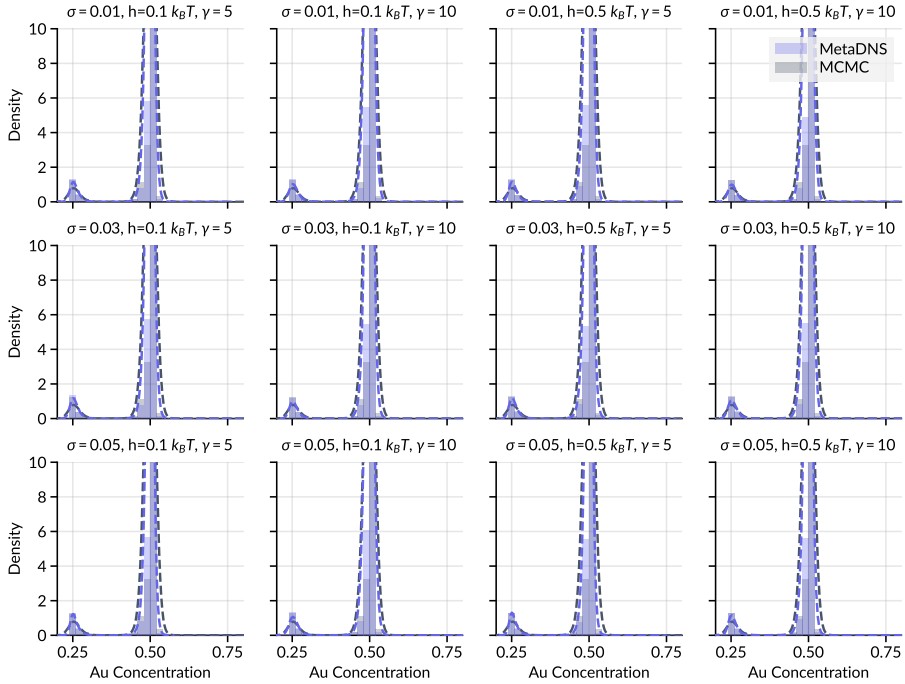

*Figure 23.* Au concentration distributions sensitivity analysis for the $4 \times 4 \times 4$ Cu-Au alloy at 500K.

## G.4. Additional Benchmark Tables

*Table 5.* Ising model results at $L = 4$.

| Inv. Temp. | Method | Mag. | Corr. | NESS ↑ | E. JS Div. ↓ | $x_\uparrow$ JS Div. ↓ |
|---|---|---|---|---|---|---|
| | MDNS | 0.472 | **0.369** | **0.985** | $1.2 \times 10^{-3}$ | $\mathbf{3.6 \times 10^{-3}}$ |
| $\beta_{\text{high}} = 0.28$ | MetaDNS | **0.473** | 0.373 | 0.960 | $\mathbf{8.4 \times 10^{-4}}$ | $4.0 \times 10^{-3}$ |
| | SW (ground truth) | 0.474 | 0.367 | / | / | / |
| | MDNS | **0.842** | 0.782 | **0.986** | $4.9 \times 10^{-4}$ | $\mathbf{1.1 \times 10^{-3}}$ |
| $\beta_{\text{crit}} = 0.4407$ | MetaDNS | 0.847 | **0.786** | 0.972 | $\mathbf{2.3 \times 10^{-4}}$ | $1.5 \times 10^{-3}$ |
| | SW (ground truth) | 0.838 | 0.787 | / | / | / |
| | MDNS | 0.976 | **0.958** | **0.994** | $1.3 \times 10^{-3}$ | $\mathbf{1.8 \times 10^{-3}}$ |
| $\beta_{\text{low}} = 0.6$ | MetaDNS | **0.974** | 0.956 | 0.924 | $\mathbf{1.1 \times 10^{-3}}$ | $2.1 \times 10^{-3}$ |
| | SW (ground truth) | 0.974 | 0.957 | / | / | / |

*Table 6.* Ising model results at $L = 8$.

| Inv. Temp. | Method | Mag. | Corr. | NESS ↑ | E. JS Div. ↓ | $x_\uparrow$ JS Div. ↓ |
|---|---|---|---|---|---|---|
| | MDNS | **0.241** | 0.320 | 0.964 | $6.7 \times 10^{-3}$ | $\mathbf{7.6 \times 10^{-3}}$ |
| $\beta_{\text{high}} = 0.28$ | MetaDNS | 0.242 | **0.324** | **0.968** | $\mathbf{6.1 \times 10^{-3}}$ | $8.4 \times 10^{-3}$ |
| | SW (ground truth) | 0.240 | 0.324 | / | / | / |
| | MDNS | 0.782 | **0.748** | **0.960** | $\mathbf{3.2 \times 10^{-3}}$ | $\mathbf{1.1 \times 10^{-2}}$ |
| $\beta_{\text{crit}} = 0.4407$ | MetaDNS | **0.779** | **0.748** | 0.940 | $7.1 \times 10^{-3}$ | $1.8 \times 10^{-2}$ |
| | SW (ground truth) | 0.775 | 0.739 | / | / | / |
| | MDNS | **0.975** | **0.956** | **0.994** | $\mathbf{1.8 \times 10^{-3}}$ | $2.1 \times 10^{-1}$ |
| $\beta_{\text{low}} = 0.6$ | MetaDNS | 0.974 | **0.956** | 0.816 | $4.5 \times 10^{-2}$ | $\mathbf{2.0 \times 10^{-2}}$ |
| | SW (ground truth) | 0.976 | 0.957 | / | / | / |

*Table 7.* Potts ($q = 3$) model results at $L = 4$. CV JS Div. is obtained by projecting samples into the 2D CV space and calculating the JS divergence vs. SW based on the obtained distribution.

| Inv. Temp. | Method | Mag. | Corr. | NESS ↑ | E. JS Div. ↓ | CV JS Div. ↓ |
|---|---|---|---|---|---|---|
| | MDNS | **0.540** | **0.131** | **0.987** | $\mathbf{2.0 \times 10^{-3}}$ | $2.5 \times 10^{-3}$ |
| $\beta_{\text{high}} = 0.5$ | MetaDNS | 0.544 | 0.134 | 0.943 | $2.7 \times 10^{-3}$ | $\mathbf{1.8 \times 10^{-3}}$ |
| | SW (ground truth) | 0.541 | 0.132 | / | / | / |
| | MDNS | 0.897 | 0.521 | **0.961** | $\mathbf{2.1 \times 10^{-3}}$ | $2.4 \times 10^{-3}$ |
| $\beta_{\text{crit}} = 1.005$ | MetaDNS | **0.895** | **0.520** | 0.853 | $3.8 \times 10^{-3}$ | $\mathbf{1.8 \times 10^{-3}}$ |
| | SW (ground truth) | 0.891 | 0.516 | / | / | / |
| | MDNS | 0.966 | 0.611 | **0.966** | $\mathbf{4.0 \times 10^{-3}}$ | $2.0 \times 10^{-3}$ |
| $\beta_{\text{low}} = 1.2$ | MetaDNS | **0.968** | **0.615** | 0.805 | $5.8 \times 10^{-3}$ | $\mathbf{9.8 \times 10^{-4}}$ |
| | SW (ground truth) | 0.970 | 0.618 | / | / | / |

*Table 8.* Potts ($q = 3$) model results at $L = 8$.

| Inv. Temp. | Method | Mag. | Corr. | NESS ↑ | E. JS Div. ↓ | CV JS Div. ↓ |
|---|---|---|---|---|---|---|
| | MDNS | **0.433** | 0.126 | **0.955** | $\mathbf{4.9 \times 10^{-3}}$ | $1.0 \times 10^{-2}$ |
| $\beta_{\text{high}} = 0.5$ | MetaDNS | 0.434 | **0.128** | 0.880 | $5.4 \times 10^{-3}$ | $\mathbf{5.9 \times 10^{-3}}$ |
| | SW (ground truth) | 0.433 | 0.128 | / | / | / |
| | MDNS | 0.849 | 0.492 | **0.910** | $\mathbf{7.8 \times 10^{-3}}$ | $9.0 \times 10^{-3}$ |
| $\beta_{\text{crit}} = 1.005$ | MetaDNS | **0.850** | **0.494** | 0.762 | $2.9 \times 10^{-2}$ | $\mathbf{4.9 \times 10^{-3}}$ |
| | SW (ground truth) | 0.853 | 0.496 | / | / | / |
| | MDNS | **0.969** | 0.615 | **0.954** | $\mathbf{2.9 \times 10^{-3}}$ | $3.2 \times 10^{-1}$ |
| $\beta_{\text{low}} = 1.2$ | MetaDNS | 0.969 | 0.616 | 0.552 | $5.3 \times 10^{-2}$ | $\mathbf{2.6 \times 10^{-3}}$ |
| | SW (ground truth) | 0.969 | 0.615 | / | / | / |

*Table 9.* Potts ($q = 3$) model results at $L = 16$.

| Inv. Temp. | Method | Mag. | Corr. | NESS ↑ | E. JS Div. ↓ | CV JS Div. ↓ |
|---|---|---|---|---|---|---|
| | MDNS | **0.383** | 0.127 | **0.925** | $9.4 \times 10^{-2}$ | $2.0 \times 10^{-2}$ |
| $\beta_{\text{high}} = 0.5$ | MetaDNS | **0.383** | 0.128 | 0.781 | $\mathbf{8.9 \times 10^{-2}}$ | $\mathbf{7.5 \times 10^{-3}}$ |
| | SW (ground truth) | 0.381 | 0.129 | / | / | / |
| | MDNS | **0.786** | 0.467 | **0.552** | $\mathbf{1.9 \times 10^{-1}}$ | $2.0 \times 10^{-2}$ |
| $\beta_{\text{crit}} = 1.005$ | MetaDNS | 0.804 | 0.476 | 0.490 | $2.3 \times 10^{-1}$ | $\mathbf{2.0 \times 10^{-2}}$ |
| | SW (ground truth) | 0.787 | 0.465 | / | / | / |
| | MDNS | **0.969** | 0.615 | **0.939** | $7.6 \times 10^{-2}$ | $3.2 \times 10^{-1}$ |
| $\beta_{\text{low}} = 1.2$ | MDNS (warm-start) | 0.922 | 0.552 | 0.016 | $4.3 \times 10^{-1}$ | $4.8 \times 10^{-1}$ |
| | MetaDNS | **0.969** | **0.615** | 0.218 | $1.6 \times 10^{-1}$ | $\mathbf{4.0 \times 10^{-3}}$ |
| | SW (ground truth) | 0.968 | 0.613 | / | / | / |

*Table 10.* Cu-Au alloy results at $2 \times 2 \times 4$ supercell. $x_{Au}$ is the gold element fraction. $x_{Au}$ JS Div. refers to the gold element fraction JS divergence between results from the neural sampler vs. MCMC.

| Temperature | Method | $x_{Au}$ | NESS ↑ | E. JS Div. ↓ | $x_{Au}$ JS Div. ↓ |
|---|---|---|---|---|---|
| | MDNS | 0.478 | **0.968** | $\mathbf{2.4 \times 10^{-3}}$ | $\mathbf{2.3 \times 10^{-3}}$ |
| 1200 K | MetaDNS | **0.482** | 0.903 | $4.6 \times 10^{-3}$ | $2.7 \times 10^{-3}$ |
| | MCMC (ground truth) | 0.487 | / | / | / |
| | MDNS | 0.437 | **0.846** | $\mathbf{1.8 \times 10^{-3}}$ | $\mathbf{3.2 \times 10^{-3}}$ |
| 680 K | MetaDNS | **0.439** | 0.830 | $1.7 \times 10^{-2}$ | $5.4 \times 10^{-3}$ |
| | MCMC (ground truth) | 0.441 | / | / | / |
| | MDNS | 0.435 | **0.939** | $\mathbf{1.7 \times 10^{-3}}$ | $3.1 \times 10^{-3}$ |
| 500 K | MetaDNS | **0.441** | 0.850 | $1.7 \times 10^{-2}$ | $\mathbf{2.5 \times 10^{-3}}$ |
| | MCMC (ground truth) | 0.441 | / | / | / |

*Table 11.* Cu-Au alloy results at $4 \times 4 \times 4$ supercell.

| Temperature | Method | $x_{Au}$ | NESS ↑ | E. JS Div. ↓ | $x_{Au}$ JS Div. ↓ |
|---|---|---|---|---|---|
| | MDNS | **0.494** | **0.787** | $\mathbf{1.1 \times 10^{-2}}$ | $\mathbf{4.3 \times 10^{-3}}$ |
| 1200 K | MetaDNS | 0.493 | 0.517 | $3.6 \times 10^{-2}$ | $5.7 \times 10^{-3}$ |
| | MCMC (ground truth) | 0.496 | / | / | / |
| | MDNS | **0.464** | **0.402** | $4.2 \times 10^{-2}$ | $\mathbf{1.6 \times 10^{-2}}$ |
| 680 K | MetaDNS | 0.465 | 0.169 | $\mathbf{1.6 \times 10^{-2}}$ | $2.1 \times 10^{-2}$ |
| | MCMC (ground truth) | 0.454 | / | / | / |
| | MDNS | 0.499 | **0.976** | $1.3 \times 10^{-1}$ | $1.3 \times 10^{-1}$ |
| | MDNS (warm-start) | **0.491** | 0.914 | $8.6 \times 10^{-2}$ | $8.9 \times 10^{-2}$ |
| 500 K | MetaDNS | 0.489 | 0.321 | $\mathbf{7.9 \times 10^{-2}}$ | $\mathbf{8.5 \times 10^{-2}}$ |
| | MCMC (ground truth) | 0.490 | / | / | / |

*Table 12.* Training and inference cost: MetaDNS vs. MCMC-based WT-MetaD for the $16 \times 16$ Potts ($q = 3$) model. Potts energies are cheap and batched on GPU, so WT-MetaD reaches shorter wall-clock training than MetaDNS despite more bias deposition steps, whereas MetaDNS pays for inner-loop neural network optimization. Training steps and wall times are consistent across temperatures. MetaDNS inference uses autoregressive forward passes; WT-MetaD inference runs MCMC under the converged static bias. Underlined method name indicates our method (MetaDNS).

| | Training (128 batch size) | | Inference (10k samples) | |
|---|---|---|---|---|
| Method | Bias Deposition Steps | Wall Time | Inference Steps | Wall Time |
| MetaDNS | 50k | 20 h | 256 | <1 min |
| WT-MetaD | 125k | 1 h | 100k | ≈30 min |

*Table 13.* Training and inference cost: MetaDNS vs. MCMC-based WT-MetaD for the $4 \times 4 \times 4$ Cu-Au alloy at 500 K (low temperature). Cluster expansion energies are expensive and sequential on CPU so training wall times are then comparable, with MetaDNS slightly faster while using fewer bias deposition steps.

| | Training (128 batch size) | | Inference (10k samples) | |
|---|---|---|---|---|
| Method | Bias Deposition Steps | Wall Time | Inference Steps | Wall Time |
| MetaDNS | 20k | 1.5 h | 64 | <1 min |
| WT-MetaD | 40k | 1.75 h | 1k | ≈40 min |

