# OpenReview forum: "MetaDNS: Enhancing Exploration in Discrete Neural Samplers via Metadynamics"
_ICML.cc/2026/Conference — ICML 2026 regular_

### Official Review · Reviewer_rrqa · 2026-02-18

**Soundness:** 2
**Presentation:** 4
**Significance:** 4
**Originality:** 3
**Overall Recommendation:** 5
**Confidence:** 5

**Summary:**

The main aim of the paper is to implement Metadynamics within the framework of neural samplers in discrete configuration spaces. The central issue with state-of-the-art methods is mode collapse: in a collective variable (CV)-based setup, the sampled full configurations fail to properly explore the free energy landscape. A classical strategy to address this limitation is Metadynamics, which the authors incorporate into the neural sampler framework in order to enhance its strengths.
From a theoretical standpoint, the work builds upon existing results, placing the contribution primarily in an empirical direction. The core results of the paper are therefore algorithmic and experimental: the proposed method is benchmarked against Swendsen–Wang for the Ising and Potts models, and against standard MCMC for Cu–Au systems, with the authors claiming a reduced computational cost.

**Compliance With Llm Reviewing Policy:**

Affirmed.

**Final Justification:**

The authors addressed all my residual concerns, hence I raise the score to Accept.

**Key Questions For Authors:**

1. Computational cost – training loop accounting

The claim of improved computational efficiency is central to the paper. Could the authors provide a detailed breakdown of the computational cost of Algorithm 1, explicitly accounting for the training loop? In particular, how does the cost of network updates (forward/backward passes, parameter optimization steps) compare to the cost of energy evaluations, and how is this incorporated into the complexity analysis presented in Figure 4?

2. Computational cost – fair comparison with MCMC and parallel methods

In the comparison with standard MCMC methods (e.g., Swendsen–Wang), the cost seems to be measured mainly in terms of energy evaluations. How would the conclusions change if wall-clock time or total FLOPs—including both sampling and training phases—were considered? Additionally, have the authors considered benchmarking against parallel tempering or other advanced MCMC schemes to provide a more comprehensive comparison with state-of-the-art sampling strategies?

3. Literature positioning

The interaction between neural networks and enhanced sampling methods (including Metadynamics and related control-based approaches) has been explored in several recent works. Could the authors clarify how their contribution differs conceptually and algorithmically from existing approaches in this direction, and expand Section 2.3 to better position the method within the broader literature on ML-enhanced sampling?

Clarifying this few points would easily move my overall evaluation to Acceptance.

**Limitations:**

Yes

**Strengths And Weaknesses:**

Soundness:

The paper is overall very sound, but in my opinion there are a few key points that require clarification. The main claim concerns computational efficiency, and Figure 4 is intended to support this statement. However, the RMSE criterion is only briefly mentioned in the main text and would benefit from a clearer explanation.
More importantly, the authors argue for a reduced computational cost primarily in terms of energy evaluations, as typically done for standard MCMC methods. It is not entirely clear, however, where the cost of the training loop is taken into account in this comparison. Since computational efficiency is a central claim of the paper, I strongly encourage the authors to include a dedicated paragraph explicitly detailing how the cost of each operation in Algorithm 1 is accounted for—clarifying, for instance, which steps are assumed to be O(1)—and to provide a transparent comparison with standard MCMC approaches.
I raise this point because, in other works on neural sampling (e.g., https://arxiv.org/pdf/2506.05231), a major computational bottleneck arises precisely from the training loop. For the comparison to be fully convincing and fair, it is important to clearly incorporate this cost into the overall complexity analysis. Moreover, a comparison with parallel tempering could also increase the soundness of the experimental results against SOTA sampling methods.

Presentation:

I find the presentation of the topic to be clear and well structured. I do not have any specific suggestions regarding the overall organization of the manuscript.

Significance:

The overall aim of the paper is very significant, since it makes sense to include Metadynamics in neural sampling routine, with the final objective to improve sampling techniques with hybrid methods (ML and MCMC). The experiments are also sufficiently simple in the first part to understand the problem, and sufficiently hard regarding Au-Cu, to hint for scalability.

Originality:

Regarding originality, I think the authors should take some more time to explore the existing literature on the interaction between neural networks and Metadynamics in sampling. After a short research, I can already ping some papers that may be relevant:
- E. Ribera Borrell, J. Quer, L. Richter, and C. Schütte, Improving control based importance sampling strategies for metastable diffusions via adapted metadynamics, SIAM J. Sci. Comput., 46(2), S298–S323 (2023).

- Du, X., Liu, S., & Gomez-Bombarelli, R. Scaling autoregressive models for lattice thermodynamics. In AI for Accelerated Materials Design-NeurIPS 2024.

- Zhang, J., Yang, Y. I., & Noé, F. (2019). Targeted adversarial learning optimized sampling. The journal of physical chemistry letters, 10(19), 5791-5797.

- Zhu, K., Trizio, E., Zhang, J., Hu, R., Jiang, L., Hou, T., & Bonati, L. (2025). Enhanced sampling in the age of machine learning: algorithms and applications. Chemical Reviews, 126(1), 671-713.

In particular, I feel Section 2.3. should be populated with more methods that may have developed related ideas.

---

> ### Author Rebuttal · Authors · 2026-03-31
>
> We thank the Reviewer for the thoughtful feedback and favorable assessment.
>
> ## W1/Q1/Q2 Computational cost
>
> We agree that computational efficiency claims need to clearly separate energy evaluations from neural network training cost and wall-clock time, and we will update the manuscript accordingly.
>
> **RMSE criterion:** The convergence panels in Figures 4 and 5 measure RMSE between the running PMF estimate and a reference PMF (the final WT-MetaD profile at 125k steps for Potts, 40k for Cu-Au). Before computing RMSE, we vertically align the two curves by shifting so their minima coincide, since the PMF is only defined up to a constant. RMSE is then computed over all discrete CV grid points in units of $k_BT$. For Potts, this is done on the full 2D CV grid; the 1D slice in the figure is for visualization only. We will add a formal definition to Appendix C.
>
> **Training loop cost:** We will clarify in Section 4.2 that the RMSE curves are mapped over bias deposition steps (outer loop cost) rather than energy evaluation steps. Each MetaDNS outer loop step has multiple inner loop steps (energy evaluation and neural network training), 5 for Potts and Cu-Au (we will update Tables 3 & 4 to better reflect these values). The baseline WT-MetaD also requires multiple MCMC steps per bias deposition, 64 for Potts and Cu-Au (Appendix F.3).
>
> **Wall-clock comparison:** Whether MetaDNS or WT-MetaD is faster in wall time depends on how expensive the energy function is. For Potts, energies are cheap and batched, so WT-MetaD finishes training in about 1 hour while MetaDNS takes roughly 20 hours due to parameter optimization overhead. For Cu-Au, the cluster expansion is expensive and sequential on CPU, so training wall times are comparable. MetaDNS is actually slightly faster (1.5 h vs 1.75 h) despite the neural network overhead. After training, MetaDNS has a large inference advantage in both cases: generating 10k samples takes under 1 minute via autoregressive forward passes, versus 30-40 minutes for WT-MetaD which must run multiple MCMC steps. We will include these wall time tables in the Appendix and discuss the tradeoff explicitly in the Discussion section. We will also add a paragraph in Section 4.2 to clarify how the relative costs of energy evaluation and neural network training affect the final wall-clock times.
>
> **Parallel tempering:** We chose WT-MetaD rather than parallel tempering (PT) as our MCMC baseline because PT operates in a fundamentally different regime: it maintains multiple replicas of fixed Boltzmann distributions, multiplying the energy budget by the number of replicas, and does not directly produce a PMF along a collective variable without additional post-processing. WT-MetaD and MetaDNS both use a single chain on a biased, evolving target, making the comparison apples-to-apples in terms of energy budget and output. For Ising and Potts, our ground truth is Swendsen–Wang, which is itself a highly efficient cluster MCMC method.
>
> ## W2/Q3 Literature positioning
>
> We thank the reviewer for the references and apologize for the omissions. We will update Sections 2.1 and 2.3 to reflect these works. MetaDNS can be placed within a broader family of ML-enhanced sampling methods that either (i) learn controls or biases in continuous space (often with MD/MCMC), or (ii) build neural samplers for discrete lattice thermodynamics, typically without explicit metadynamics. These works are relevant context but do not change the originality or conclusions of the paper, since none of them integrate well-tempered metadynamics into discrete neural samplers or directly address mode collapse and barrier crossing in discrete configurational spaces.
>
> **Continuous-space control and bias learning (Section 2.3):** Ribera Borrell et al. (2024) combine stochastic optimal control-based importance sampling with adaptive metadynamics; this can be seen as a precursor in the SOC line to the WT-ASBS work of Nam et al. (2025), already cited. Zhang et al. (2019) propose a GAN-style framework to learn an optimal bias and transport plan that lowers free-energy barriers in molecular settings. Zhu et al. (2025) review how ML integrates with enhanced sampling through learned CVs, improved biasing, and generative strategies. These methods all operate in continuous/molecular settings, whereas MetaDNS targets discrete configuration spaces (Ising, Potts, alloys) and couples WT-MetaD directly to the training dynamics of discrete neural samplers.
>
> **Neural samplers for lattice thermodynamics (Section 2.1):** Du et al. (2024) scale any-order autoregressive models to larger lattices, building on the already-cited Damewood et al. (2022) and Liu et al. (2024). However, these methods train directly against the target Hamiltonian and do not incorporate metadynamics or history-dependent biasing, nor do they claim to address mode collapse or barrier crossing.

---

> > ### Author Rebuttal · Reviewer_rrqa · 2026-03-31
> >
> > Thank you for the clarification and the response.
> >
> > I am perfectly fine to raise the score to accept, but would be nice to link (as allowed by ICML rules) some novel plot and tables summarising the comment on the computational cost. Thanks.

---

> > > ### Author Response · Authors · 2026-03-31
> > >
> > > Thank you for the positive feedback. Here's the link to our new tables summarizing computational cost comparison: https://imgur.com/a/61FuaI7

---

### Official Review · Reviewer_xM3R · 2026-03-01

**Soundness:** 2
**Presentation:** 3
**Significance:** 2
**Originality:** 3
**Overall Recommendation:** 4
**Confidence:** 3

**Summary:**

This paper proposes *MetaDNS*, a framework to improve exploration for *discrete neural samplers* targeting Boltzmann distributions over discrete configurations. The key idea is to integrate *well-tempered metadynamics* into training by maintaining a *history-dependent bias potential* over user-chosen low-dimensional *collective variables (CVs)*. The method alternates between (i) training a neural sampler to match a *biased* distribution defined by the original energy plus the current bias, and (ii) updating the bias by depositing “hills” in CV space at visited regions, with a well-tempered schedule to prevent unbounded bias growth. At inference/evaluation time, the method uses *importance reweighting* (bias-based and/or likelihood-based when tractable) to estimate observables under the original unbiased target. Experiments on Ising and Potts models and a Cu–Au alloy benchmark aim to show improved mode coverage, access to barrier/intermediate states enabling free-energy reconstruction, and reduced energy-evaluation cost compared with MCMC-based metadynamics baselines.

**Compliance With Llm Reviewing Policy:**

Affirmed.

**Final Justification:**

The rebuttal addressed my concerns. I adjust my score accordingly

**Key Questions For Authors:**

- Could the authors provide systematic sensitivity studies (even limited) showing how results change across reasonable ranges of the used hyperparameters and different CV discretizations in MetaDNS?
- Could the authors comment baselines that adaptively modify temperature (or use parallel/simulated tempering) or otherwise increase barrier-crossing probability (also see Weaknesses third point)? Experimental comparisons between such a baseline and MetaDNS would strengthen the soundness of the proposed method.

**Limitations:**

yes

**Strengths And Weaknesses:**

**Strengths**

- The paper tackles a relevant and well-motivated problem: *exploration and mode coverage* in discrete Boltzmann sampling, especially in *low-temperature, high-barrier* regimes where standard MCMC can mix slowly and neural samplers may collapse.
- The overall framework is conceptually clear: it combines a classical enhanced-sampling technique (WT-MetaD) with modern discrete neural samplers, and the algorithmic structure (inner loop fitting + outer loop bias update) is easy to follow.
- The experiments span both pedagogical physics benchmarks (Ising/Potts) and a more realistic materials setting (Cu–Au), which helps demonstrate potential practical value. The discussion of reweighting options (bias-based vs likelihood-based where possible) is also useful.

**Weaknesses**

Overall, I believe the general math framework sounds solid. My major concerns lie in whether this framework can actually benefit overcome energy barriers in real-world applications.

- From my understanding, MetaDNS deliberately biases the sampling distribution to encourage exploration, then relies on importance reweighting to recover unbiased estimates. This design may introduce *a systemic tension*: aggressive flattening tends to increase weight variance (low ESS/NESS), potentially making estimation noisy and unstable. The authors mention approaches to mitigations this (e.g., inner-loop training, well-tempered hill deposition, optional corrections), but the presented evidence feels more like *ad hoc alleviation* than a systematic demonstration that the approach is stable across regimes.
- Additionally, this introduces additional metadynamics-specific hyperparameters (e.g., hill height/width and bias factor, etc.) and depends on *CV design*. The paper acknowledges sensitivity, but does not provide a systematic sensitivity study. In practice, this makes the method harder to adopt: users may not know how to tune these knobs, and tuning can be expensive precisely in the settings where energy evaluations are costly.
- The authors mention that collapse happens mainly at low temperature/high barriers, and it seems that MDNS works fine at higher temperatures (in Table 1 / Figure 2). I believe a more natural and simpler baseline for barrier crossing is *adaptive tempering/temperature schedules / step-size adaptations* (or related tempering strategies). Specifically, when encountering energy barrier, we adaptivaly adjust temperature / step size to escape the barrier. However, I did not see a comparison against such approaches, which makes it hard for me to identify whether MetaDNS is necessary or whether a simpler baseline would address the same issue with fewer assumptions/engineering efforts.
- The current evaluation does not convincingly establish that it is the best tradeoff among (i) exploration quality, (ii) unbiasedness/stability of estimates, and (iii) tuning/engineering effort. The gains are clearest relative to MCMC-based WT-MetaD in some settings, but the broader claim of solving mode collapse for discrete neural samplers would benefit from stronger and more comprehensive comparisons.

---

> ### Author Rebuttal · Authors · 2026-03-31
>
> We thank the Reviewer for their assessment.
>
> ## W1
> Thank you for raising this point. We think the apparent tension here is largely a diagnostic artifact. MetaDNS is not designed to maximize the ESS of raw importance weights $\exp(\beta V)$ or to keep the proposal close to the target at all times. Its goal is to minimize the error of Monte Carlo estimates under a fixed sample budget, especially in rare or barrier regions. In those regimes, an exact unbiased sampler already performs poorly: if a region has probability mass $p(s)$, then $N$ unbiased samples visit it only $N p(s)$ times, so the relative error scales as $O((N p(s))^{-1/2})$, which is prohibitive when $p(s)$ is exponentially small. Well-tempered flattening deliberately reallocates samples toward such regions. As a result, a global ESS/NESS based on raw weights can decrease while the variance of the final reweighted estimator decreases as well. For this reason, low ESS by itself is not a reliable indicator of instability in our setting.
>
> Importantly, the correction in MetaDNS is not generic high-variance full-state importance sampling. The bias acts only through the low-dimensional statistic $s$, and at the well-tempered fixed point the marginal is controlled as $p_V(s)\propto p(s)^{1/\gamma}$. This gives a systematic exploration versus reweighting tradeoff, rather than an ad hoc one, and makes the weights substantially better behaved than full-configuration reweighting. The inner-loop training and hill-deposition schedule are mechanisms to realize this controlled flattened marginal stably, not patchwork fixes for an unstable method.
>
> ## W2/Q1
>
> We performed a sensitivity analysis of bias width $\sigma$, bias height $h$, and bias factor $\gamma$ for the most challenging systems at low temperatures, 16x16 Ising, 16x16 Potts, and 4x4x4 Cu-Au, with all other training hyperparameters and the number of training steps fixed. Additionally for the 16x16 Ising model, we did a sensitivity analysis for the number of CV bins $\in \{129, 257\}$
> - $\sigma \in \{0.01, 0.03, 0.05\}$
> - $h \in \{0.1 k_\text{B}T, 0.5 k_\text{B}T\}$
> - $\gamma \in \{5, 10\}$
>
> In most of the metadynamic settings, MetaDNS successfully recovered all modes and achieved strong agreement of observables that are consistent with the reference.
> NESS is higher for lower $\gamma$, which is as we expect, since that means lower deviation of the sampled distribution from the true target distribution.
> 1. For 16x16 Ising, NESS ranged from 0.30 to 0.60, consistent with the low-temperature MetaDNS value in Table 1. Mode collapse was observed for the extreme case of lowest $\sigma, h, \gamma$ for both CV bin sizes, with a larger deviation of the observed free energy and an artificially high NESS. The likely explanation is that these set of hyperparams makes building up the bias very slow and a fixed amount of 50k steps might be insufficient for the sampler to walk out of the existing mode.
> 2. For 16x16 Potts, NESS ranged from 0.20 to 0.50, consistent with the low-temperature MetaDNS value in Table 9. For Potts, mode collapse was observed for the extreme case of lowest $\sigma, h, \gamma$ as well.
> 3. For 4x4x4 Cu-Au, NESS ranged from 0.20 to 0.40, consistent with the low-temperature MetaDNS value in Table 11.
>
> Overall, these experiments show that MetaDNS remains effective without careful tuning, therefore making it more accessible to users. We will show these plots in the final version of the manuscript.
>
> ## W3/Q2
>
> In our Q1 response to Reviewer nTuR, we demonstrate how a temperature-modification-based method could fail. Additionally, such a trained neural sampler still samples just the low-energy regions and might not pursue our goal of sampling the intermediate, high-energy states that are exponentially suppressed under the Boltzmann distribution.
>
> ## W4
>
> We address the three dimensions in turn. A clarification on scope: we do not claim to have "solved mode collapse" but our listed contributions are bounded and supported by our evaluation metrics.
> 1. Exploration quality. Our experiments compare MetaDNS against MDNS (the SOTA discrete neural sampler), WT-MetaD (the classic enhanced sampling method), and Swendsen–Wang/long MCMC as ground truth. On mode coverage, MetaDNS successfully recovers all modes across low-temperature Ising, Potts, and Cu-Au benchmarks that MDNS misses entirely. On free energy estimation, MetaDNS produces PMFs at low temperature while MDNS is unable to.
> 2. Unbiasedness/stability. Please see our response to W1.
> 3. Tuning/engineering effort. The additional hyperparameters relative to MDNS are exactly the standard WT-MetaD parameters ($\gamma$, $\sigma$, $h$) and we empirically show that they do not require careful tuning to work (see our response to W2/Q1). The engineering overhead of MetaDNS is therefore modest relative to the capabilities it adds.

---

> > ### Author Rebuttal · Reviewer_xM3R · 2026-04-02
> >
> > I appreciate the authors' detailed response, which addressed my concerns. I adjusted my score accordingly.

---

> > > ### Author Response · Authors · 2026-04-03
> > >
> > > Thank you for the positive feedback.

---

### Official Review · Reviewer_TRpZ · 2026-03-07

**Soundness:** 3
**Presentation:** 3
**Significance:** 4
**Originality:** 3
**Overall Recommendation:** 5
**Confidence:** 4

**Summary:**

This work proposes to apply enhanced sampling methods, based on collective variables, to enhance discrete neural sampler training, which can be seen as a discrete variant on the concurrent work [1]. The proposed framework is universal to any discrete neural sampler training, as the method is adding bias potential to the target potential to fast step out of the slow-modes, therefore the author mainly experiments on employing their framework to the SOTA discrete neural samplers, and compare to the well-tempered Metadynamics.

The author conducts experiments on a variety of benchmarks, showcasing the effectiveness of the proposed framework. Claims in the manuscript are justified. While the main issue is the lack of clarity regarding several techniques used in the paper: (i) the details of the base neural sampler should be stated in the appendix; (ii) since the RND-based resampling is mentioned, it would be better to extend the current appendix B with at least a pseudo-code/algorithm. These contents might be clear without stating for people working in relevant regions, but seems not friendly to other people; (iii) it is unclear that how different the proposed reweighing schemes are in practice.

Overall, despite the suggestion on clarifying the paper, the soundness and positive numerical results suggest the review to recommend an accept.

[1] Nam, Juno, et al. "Enhancing diffusion-based sampling with molecular collective variables." arXiv preprint arXiv:2510.11923 (2025).

**Compliance With Llm Reviewing Policy:**

Affirmed.

**Final Justification:**

Two of my concerns, regarding the reweighting scheme and additional details for MDNS, are solved. Therefore, I will maintain my score as a clear accept.

**Key Questions For Authors:**

**1. Clarification on the MDNS and Reweighting Strategies**
Could you elaborate on the specific details of the MDNS training pipeline? Since MDNS appears to be a non-exact density sampler, it seems that either Bias-based Reweighting or RND Reweighting could be applied. Could you clarify exactly which reweighting scheme is used in practice? Furthermore, I would appreciate an empirical comparison or discussion detailing the performance differences between using these two different reweighting strategies within your framework.

* **How this impacts my evaluation:** The current presentation leaves some ambiguity regarding the reproducibility of the MDNS pipeline and the justification for the chosen reweighting scheme. Providing these details and a brief ablation study on the reweighting strategies would resolve these confusing points, strengthen my confidence in the paper's methodological soundness, and likely improve my score for the Presentation and Soundness dimensions.

**Limitations:**

Yes

**Strengths And Weaknesses:**

# Soundness
The methodology employed in this work is technically sound. Metadynamics has a well-established history in the field of enhanced sampling, and its application within this specific context is methodologically appropriate and rigorous. The theoretical foundations of the approach are solid.

# Presentation
Overall, the submission is clearly written and the general narrative is easy to follow.  However, the paper would benefit from additional detail in a few key areas:
* **The MDNS Training Pipeline:** A more comprehensive breakdown of the training process is needed in the appendix.
* **Reweighing Schemes:** The authors should clarify exactly which reweighing scheme is utilized in practice. Furthermore, it would be highly beneficial to include a discussion or empirical comparison of this chosen scheme against alternative reweighing methods to justify its selection.

# Significance
The paper addresses the highly relevant and important problem of sampling from discrete distributions, which is important in statistics and physics. The proposed framework is shown to effectively improve sampling efficiency, indicating that this work provides clear practical utility and could easily influence future applications in the field.

# Originality
The contributions of this work are sufficiently novel. It introduces a creative combination of existing ideas by being the first to apply Metadynamics to discrete neural sampler training. While the authors appropriately position their work by acknowledging concurrent research that explores a similar scheme in continuous spaces, extending this methodology to discrete spaces and achieving improvement is a great contribution to the community.

---

> ### Author Rebuttal · Authors · 2026-03-31
>
> We thank the Reviewer for the positive assessment and the recommendation to accept. We will add a pseudocode of the MDNS training pipeline in the Appendix. MDNS can can be interpreted as an exact density sampler, as we document in Appendix B. We will make this clearer in the final version (we are not allowed to upload a revised version during rebuttals.) In our work, we found global metrics such as energy and magnetization observables to be sufficient with bias-based reweighting while for local metrics such as two-point correlation, likelihood-based reweighting was necessary. We will add additional figures comparing two-point correlation betwen bias-based reweighting and likelihood-based reweighting.

---

> > ### Author Rebuttal · Reviewer_TRpZ · 2026-03-31
> >
> > Thank you for clarifying. I don't have any other questions and will maintain my score.

---

### Official Review · Reviewer_nTuR · 2026-03-12

**Soundness:** 2
**Presentation:** 3
**Significance:** 1
**Originality:** 3
**Overall Recommendation:** 4
**Confidence:** 3

**Summary:**

This paper investigates sampling discrete distributions using neural samplers, when only unnormalized densities are known. In particular, they improve upon the recent MDNS sampler, which is a discrete diffusion-based sampler for physical systems. The key innovation is to integrate Well-Tempered Metadynamics into the training of the model, to ensure the model does not get trapped in a single mode. The training alternates between an inner loop to train the sampler on the biased distribution and an outer loop that updates the bias potential by sampling from the trained model. During sampling from the final model, importance sampling is used to correct for the effect of the bias during training.

**Compliance With Llm Reviewing Policy:**

Affirmed.

**Final Justification:**

Initially, I had concerns about comparisons with MDNS and whether they were fair. During the rebuttal, the authors have addressed my main concerns and have delivered full comparisons with MDNS with warm-start.

I believe that the paper could be strengthened by more detailed experiments and application across a wider range of systems. But overall, I believe that the ideas presented in this paper are novel and meet the bar for acceptance.

**Key Questions For Authors:**

1. As I was already mentioning above, I think it is crucial that authors need to compare against MDNS with “warm-start”. What is the reasoning for not including this benchmark / this setup as a baseline / additional model? From what I can see, it seems like this warm start would essentially overcome similar issues. Are there any drawbacks I am missing? Seeing a comparison with this setup and your method actually improving on it would greatly increase my confidence in the paper.
2. It seems to me that the example in Fig 2a (mode collapse for MDNS) is a rather extreme case with essentially two delta peaks. Is this behavior something that the authors found to get worse and worse with lower temperatures? or is the specific temperature of $\beta=0.60$ a specific error case? Are these actually relevant temperatures for these systems?
3. Getting stuck in a mode can (in principle) be overcome by increasing the simulation time. For how log did the authors let these simulations run? Did they already hit practical limits (as in compute time) with MDNS or did they essentially do "early stopping"?
4. As the authors state in the limitations section, the performance of the algorithm is dependent on the hyperparameters of the bias potential. How much fine-tuning was necessary to achieve these results?

**Limitations:**

yes

**Strengths And Weaknesses:**

## Strengths
- The manuscript is well-written and easy to follow. The benchmark systems are meaningful, and the metrics they investigate are well suited.
- The combination of neural samplers with accelerated sampling techniques is a novel direction that addresses important issues and makes neural samplers more easily applicable to real-world systems.
- The method is well explained, motivated and justified. Especially the algorithm allows readers to quickly understand their method and contributions.

## Weaknesses
- My most important criticism is that the paper proposes a neural sampler that does not sample the true distribution but a biased one: the approach to train the neural sampler on a biased distribution and then correct for the bias during inference using importance sampling is rather wasteful. What you would rather want is a neural sampler with a minimal bias (for more efficient sampling) that at the same time does not suffer from mode collapse during inference. I think this could be achieved by a multi-stage training strategy, where the model is first trained using the approach presented in this paper (to discover the full configuration space) and then the bias potential is gradually turned off such that the model in the end will learn the true distribution well but with all the modes.
- In fact, the original MDNS paper already contains a similar approach to train the sampler using a “warm-up” distribution with higher temperature first (see appendix D.2.4 of the MDNS paper). I suggest the authors should benchmark against this instead of vanilla MDNS for a fairer comparison in the low temperature regime. More on that in questions.
- The performance of their algorithm at high temperatures is similar or even slightly worse than the baseline, as can be seen in Table 1. I think this could be addressed with the ideas from above.

Minor nitpick: Figure 1 is never cited.

---

> ### Author Rebuttal · Authors · 2026-03-31
>
> We thank the Reviewer for their assessment.
>
> ## W1
> We agree that, if the sole objective were to output unweighted draws from the target distribution $\pi$, then a low-bias final model would be desirable. However, our method is better understood as learning an adaptive proposal for Monte Carlo estimation, not as requiring the raw proposal itself to equal $\pi$ at every stage. In low-temperature discrete systems, mode discovery is only part of the difficulty. Let $s=\xi(x)$ be a low-dimensional summary statistic and $p(s)$ its marginal under $\pi$. Even an exact unbiased sampler visits bin $s$ only $N p(s)$ times in $N$ draws, so the relative error of estimating that region scales as $O((N p(s))^{-1/2})$. For intermediate or barrier regions, $p(s)$ is exponentially small, so unbiased sampling remains exponentially inefficient there. Recent enhanced diffusion sampling works [1, 2] make the same distinction in a continuous setting: independent sample generation can remove slow mixing, but they do not remove the rare-state problem. [1] Nam et al., 2025 WT-ASBS [2] Xie et al., arXiv:2602.16634
>
> For this reason, the bias in MetaDNS is not only a temporary device to discover modes and then discard. It is the mechanism that reallocates probability mass toward rare but inference-critical regions. As stated in Appendix A and Section 3, the well-tempered fixed point gives $V^\star(s)=-(1-1/\gamma)F(s)+c$, so the biased marginal becomes $p_V(s)\propto p(s)^{1/\gamma}$. Thus regions that are exponentially rare under $\pi$ become much more visible under the proposal, while unbiased expectations are still recovered by importance weights. This is also a form of variance reduction in a sense that the goal is not to make the proposal unbiased, but to make the estimator accurate under a finite sample budget. Because the correction acts only on the low-dimensional statistic $s$, it is also much cheaper and lower-variance than full-state importance correction.
>
> ## W2
> Warm start can indeed alleviate the mode collapse issue in some cases, as demonstrated for the 16x16 Ising model in the MDNS paper (MDNS paper Figure 7: 20k high temperature training steps followed by 30k low temperature training steps). However, the initial warm-up temperature and the number of warm-up steps are arguably hyperparameters to be tuned. Moreover, warm-up/annealing from high temperature is not guaranteed to work. We provide such an example in our response to Q1.
>
> ## W3
> At high temperature, MetaDNS compares favorably with MDNS. At critical and low temperatures, our NESS metric is undoubtly impacted since we're sampling from a biased distribution. As we argue in our response to W1, this is a necessary trade off to access samples in the intermediate, high-energy regions as well as metastable states for better evaluation of the free-energy profile.
>
> ## Q1
> The true distribution might be difficult to learn if the modes are well separated and multimodal, as in the case of 16x16 Potts q=3 model. We will show an example using the same 50k step budget for the 16x16 Potts model, with a similar 20k warm-up at high temp followed by a 30k low temp step. In this example, warm-up/annealing does not solve the mode collapse problem. The performance of the annealed MDNS model is worse than that of the directly-trained MDNS model at low temperature, even though both models discover only a single mode (out of three modes). These results also make sense since the MDNS paper required a total of 100k training steps (30k high temp, 70k low temp) for the 16x16 Potts model to achieve its reported performance. Meanwhile, our MetaDNS method is able to automatically discover all three modes in 50k steps with competitive performance w.r.t. the ground truth (Table 9).
>
> ## Q2
> This specific low temperature for Ising and Potts models were adopted from the MDNS paper. For better demonstration, we train additional 16x16 Ising models at $\beta = \{0.5, 0.7, 0.8\}$ to show that mode collapse is inevitable when trained at inverse temperatures above the critical inverse temperature, i.e., $\beta > \beta_{\text{crit}}=0.4407$.
>
> ## Q3
> We politely disgree, especially in the case where the modes are well separated. For our original 16x16 Ising runs, we set 50k steps that ran in ~4 hours on an A100 GPU. We will additionally show a 100k-step training run for a total of 8 hours which still does not avoid mode collapse. This mode collapse issue is also documented in the Guo et al., 2025 PDNS paper (follow up to MDNS).
>
> ## Q4
> We will include an additional sensitivity analysis section to show that our MetaDNS is stable over a broad range of metadynamics hyperparameters, bias width $\sigma$, bias height $h$, and bias factor $\gamma$ for the most challenging low-temperature, high dimensional systems, 16x16 Ising, 16x16 Potts, and 4x4x4 Cu-Au. Please see our W2/Q1 response to Reviewer xM3R for details.

---

> > ### Author Rebuttal · Reviewer_nTuR · 2026-04-02
> >
> > I would like to thank the authors for their response and additional experiments. However, I have some follow-up questions.
> >
> > I think the warm-start issue is the main concern I still have with the paper. So I would like to clarify a few points:
> >
> > > We will show an example using the same 50k step budget for the 16x16 Potts model, with a similar 20k warm-up at high temp followed by a 30k low temp step.
> >
> > If the authors have already performed these comparisons with warm-start, could they show some results (i.e., table or figures)? It would greatly increase my confidence in this work if I could see updated tables/figures that compare with warm-start version instead of the regular one. Please note that you are allowed to post links to those, as long as they are fully anonymized, and I would appreciate it.
> >
> > > as in the case of 16x16 Potts q=3 model
> >
> > On a similar note, it is great to see that warm-start does not fully solve all the problems. However, I would be interested in seeing a discussion for the other settings and experiments from the paper and how much they change with a warm-start. In my opinion, the warm-start option should be included for most tables.
> >
> > I understand that within this short timeframe it will not be possible to do this for all experiments, but maybe the authors could further show some results and clarifications on this. It would be greatly appreciated.

---

> > > ### Author Response · Authors · 2026-04-03
> > >
> > > Thank you for the suggestion, and apologies for not uploading these additional results earlier. We were initially unsure about the rules for adding new tables and figures during rebuttal.
> > >
> > > We now provide warm-start experiments for the key systems that face mode collapse at low temperatures (16×16 Ising, 16×16 Potts, and 4×4×4 Cu–Au), using the same total training budget as MDNS and MetaDNS in each case. The updated tables and figures are available here:
> > > https://anonymous.4open.science/r/MetaDNS_ICML2026-E661/
> > >
> > > Summary of training setup:
> > > - Ising (16×16): 20k high-T + 30k low-T (50k total steps)
> > > - Potts (16×16): 20k high-T + 30k low-T (50k total steps)
> > > - Cu–Au (4×4×4): 10k high-T + 10k low-T (20k total steps)
> > >
> > > ### Key takeaway
> > >
> > > Warm-starting can help in some cases, but it is insufficient to resolve mode collapse in more challenging regimes (e.g., Potts at low temperature). This indicates that the core issue is not simply initialization, but fundamentally about mode exploration and distributional coverage.
> > >
> > > ### Detailed observations
> > >
> > > #### Potts (Table 9, Figures 12 & 15)
> > > Warm-start fails most clearly in this setting. The warm-started MDNS model still exhibits severe mode collapse, with extremely low NESS (0.016) and degraded observables. This can be understood from the large mismatch between high- and low-temperature distributions, with very limited overlap in support (see original manuscript Figures 3a & 10). As a result, warm-starting does not provide sufficient coverage of low-temperature modes.
> > >
> > > #### Ising (Table 1)
> > > Warm-start alleviates mode collapse, but results are mixed:
> > > - Correlation is slightly worse than both MDNS and MetaDNS
> > > - Energy JS divergence increases relative to MDNS
> > > - $x_\uparrow$ JS divergence improves due to better phase coverage
> > >
> > > Overall, warm-start improves diversity but does not consistently improve accuracy.
> > >
> > > #### Cu–Au (Table 11)
> > > Warm-start leads to better agreement in both energy and composition ($x_\text{Au}$) compared to MDNS, but still underperforms MetaDNS.
> > > (We also corrected the low-temperature MCMC ground-truth value for $x_\text{Au}$ in this table.)
> > >
> > > ### Conclusion
> > >
> > > Across all systems, warm-start might provide partial but inconsistent improvements, and fails in the most challenging setting (16×16 Potts). This reinforces that its limitation is not merely initialization, but that it does not address the underlying rare-state sampling problem. When the low-temperature distribution concentrates mass in well-separated regions, high-temperature samples offer only limited coverage of these regions. As a result, warm-starting alone cannot reliably recover intermediate or low-probability states that are critical for accurate estimation.
> > >
> > > In contrast, MetaDNS directly targets this limitation by reallocating probability mass toward under-sampled regions, leading to more accurate and stable estimates under a finite sampling budget.
> > >
> > > We hope these additional experiments and clarifications help address the Reviewer’s concerns and strengthen confidence in our conclusions. We appreciate the thoughtful suggestion and attention to our work.

---

### Decision · Program_Chairs · 2026-04-30

**Decision:**

Accept (regular)

**Comment:**

The paper proposes MetaDNS, a novel integration of well-tempered metadynamics into discrete neural samplers to improve exploration in multimodal, low-temperature regimes. Reviewers generally found the method technically sound, well motivated, and potentially impactful, with strong empirical results on Ising, Potts, and Cu-Au benchmarks. The main concerns centered on fairness of comparisons, sensitivity to metadynamics hyperparameters, and the accounting of computational cost. In the rebuttal, the authors substantially addressed these issues by adding warm-start comparisons, clarifying the role of reweighting and training cost, and providing additional sensitivity and computational-cost discussion. While some limitations remain in the breadth of baselines and the practical complexity of tuning, the overall reviewer consensus after rebuttal is positive and supports acceptance.